# Conditional Synthesis of 3D Molecules with Time Correction Sampler

**Hojung Jung**[1][*] **Youngrok Park**[1][*] **Laura Schmid**[1] **Jaehyeong Jo**[1]
**Dongkyu Lee**[2] **Bongsang Kim**[2] **Se-Young Yun**[1][†] **Jinwoo Shin**[1][†]
KAIST AI[1]    LG Electronics[2]
{ghwjd7281, yr-park, yunseyoung, jinwooshin}@kaist.ac.kr

## Abstract

Diffusion models have demonstrated remarkable success in various domains, including molecular generation. However, conditional molecular generation remains a fundamental challenge due to an intrinsic trade-off between targeting specific chemical properties and generating meaningful samples from the data distribution. In this work, we present Time-Aware Conditional Synthesis (TACS), a novel approach to conditional generation on diffusion models. It integrates adaptively controlled plug-and-play "online" guidance into a diffusion model, driving samples toward the desired properties while maintaining validity and stability. A key component of our algorithm is our new type of diffusion sampler, Time Correction Sampler (TCS), which is used to control guidance and ensure that the generated molecules remain on the correct manifold at each reverse step of the diffusion process at the same time. Our proposed method demonstrates significant performance in conditional 3D molecular generation and offers a promising approach towards inverse molecular design, potentially facilitating advancements in drug discovery, materials science, and other related fields.

## 1 Introduction

Discovering molecules with specific target properties is a fundamental challenge in modern chemistry, with significant implications for various domains such as drug discovery and materials science [49, 44, 7]. While diffusion models have shown great success in the generation of real-world molecules [55, 31], their primary goal is often simply to generate realistic molecules without considering specific properties, which can lead to producing molecules with undesirable chemical properties.

Existing works address this issue by leveraging controllable diffusion frameworks to generate molecules with desired properties [23, 5]. One approach is to use classifier guidance [54], which utilizes auxiliary trained classifiers to guide the diffusion process [5]. An alternative is to use classifier-free guidance (CFG) [19], which directly trains the diffusion models on condition-labeled data. While both approaches can generate stable molecules, they struggle to generate truly desirable molecules due to the complex structures and the discrete nature of atomic features [24].

On the other hand, recent works [51, 18] have introduced training-free guidance for controllable generation in a plug-and-play manner, which we hereafter refer to as online guidance (OG). This approach can directly estimate the conditional score with unconditional diffusion model. However, our analysis shows that applying online guidance into the molecular generation can result in generating samples with significantly low molecular stability and validity due to the stepwise enforcement of specific conditions without considering the original distribution at each timestep.

---

[*]Equal contribution
[†]Corresponding authors

38th Conference on Neural Information Processing Systems (NeurIPS 2024).

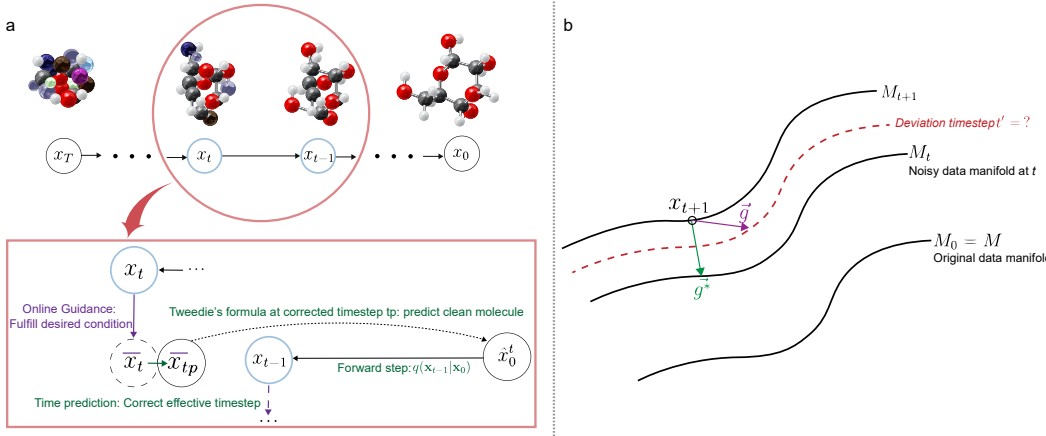

Figure 1: **(a)** Overview of Time-Aware Conditional Synthesis (TACS). TACS helps generate high-quality samples that match target condition while following basic properties of the molecules. At each timestep $t$, online guidance is applied to push $x_t$ towards the desired condition. Time Predictor finds the desired timestep $t_p$ for $x_t$ after applying the guidance. Using predicted timestep $t_p$, Tweedie's formula is used to predict the clean molecule $\hat{x}_0^t$. Finally, forward process $q(x_{t-1}|x_0)$ is applied to proceed to the next denoising step of $t-1$. **(b)** Motivation for TACS. Applying online guidance ($\vec{g}$, purple) can shift the generated samples away from the correct data manifold corresponding to the current timestep. This undesirable deviation (red) can be avoided by using time correction to first measure the deviated timestep $t'$, then adjusting the guidance to get corrected guidance vector ($\vec{g*}$, green), which keeps the generated samples stay on the correct data manifold.

To address this gap, we propose *Time-Aware Conditional Synthesis* (TACS), a novel framework for generating 3D molecules. TACS utilizes the online guidance in tandem with a novel diffusion sampling technique that we call *Time Correction Sampler* (TCS). TCS explicitly considers the possibility that online guidance can steer the generated sample away from the desired data manifold at each step of the diffusion model's denoising process, resulting in an 'effective timestep' that does not match the correct timestep during the generation. TCS then corrects this mismatch between the timesteps, thereby effectively preventing samples from deviating from the target distribution while ensuring they satisfy the desired conditions. By combining online guidance with TCS and integrating them into a diffusion model, TACS allows generated samples to strike a balance between approaching the target property and remaining faithful to the target distribution throughout the denoising process.

To the best of our knowledge, TACS is the first diffusion framework that simultaneously addresses inverse molecular design and data consistency, two critical objectives that often conflict.

We summarize our main contributions as follows:

- We propose Time-Aware Conditional Synthesis (TACS), a new framework for diffusion model that utilizes adaptively corrected online guidance during the generation.

- We introduce Time Correction Sampler (TCS), a novel diffusion sampling technique that ensures the generated samples remain faithful to the data distribution. It includes a time predictor, an equivariant graph neural network that accurately estimates the correct data manifold during inference by predicting the time information of the generation process.

- Through extensive experiments on a 3D molecule dataset, we demonstrate that TACS outperforms previous state-of-the-art methods by producing samples that closely match the desired quantum chemical properties while maintaining data consistency.

## 2   Related Works

**Diffusion models**   Diffusion models have achieved great success in a variety of domains, including generation of images [12, 46], audio generation [30], videos [21, 36, 42], and point clouds [8, 37]. A particular highlight in the success of diffusion models is their potential to generate molecules that

can form the basis of new, previously unseen, medical compounds. Multiple approaches have been explored to achieve this. For instance, graph diffusion such as GeoDiff [57], GDSS [26], and Di-Gress [54] can generate graph structures that correspond to molecular candidates. Additionally, some approaches incorporate chemical knowledge tailored to specific applications, such as RFdiffusion for protein design [55], a method based on the RoseTTAFold structure prediction network [31]. Other previous literature considers different domain-specific applications, such as diffusion for molecular docking [11], or molecular conformer generation [25]. Most relevant to our work, diffusion models have also shown promising results in synthesizing 3D molecules [23, 58, 27], generating stable and valid 3D structures. Recently, some of the works further advance 3D molecule generation techniques by making more reliable diffusion process [63] or fast generation [22] of 3D molecules.

**Conditional molecular generation**   Deep generative models [38, 23, 58] have made considerable progress in synthesizing 3D molecules with specific properties. Specifically, conditional diffusion models [23, 5, 58, 18] have achieved noticeable improvements in synthesizing realistic molecules. EDM [23] trains separate conditional diffusion models for each type of chemical condition, while EEGSDE [5] trains an additional energy-based model to provide conditional guidance during the inference. GeoLDM [58] utilizes a latent diffusion model [46] to run the diffusion process in the latent space. MuDM [18] applies online guidance to simultaneously target multiple properties. However, existing methods either produce unstable and invalid molecular structures or are unable to accurately meet the target conditions. To overcome these limitations, we propose TACS, a novel framework which ensures the generative process remains faithful to the learned marginal distributions in each timestep while effectively guiding the samples to meet the desired quantum chemical properties.

**Adaptive inference schedule**   Recent efforts have explored adaptive inference schedules to enhance the fidelity of samples generated by diffusion models with timestep information. [59] proposes adjusting the noise schedule during reverse diffusion, while [60] optimizes input timesteps for a more accurate reverse trajectory. TS-DPM [34] mitigates exposure bias by adaptively shifting timesteps during inference to align with the training variance. Similar to our time preidction mechanism, several works leverage auxilary network to classify time information with different motivation and purpose. DMCMC [28] introduce noise classifier for accelerating inference during earlier diffusion steps and MoreRed [61] leverages a timestep prediction network to initialize or adaptively guide the reverse diffusion process using the adjusted timesteps for molecular relaxation. In contrast, our TACS is specifically designed for conditional generation, predicts timesteps at each denoising step to refine clean sample estimates via Tweedie's formula, ensuring alignment with the data manifold and adherence to target conditions. Further comparison is provided in Appendix E.

## 3   Preliminaries

**Diffusion models**  Diffusion models [50, 20, 52] are a type of generative model that learn to reverse a multi-step forward noising process applied to the given data. In the forward process, noise is gradually injected into the ground truth data, $\mathbf{x_0} \sim q_0$, until it becomes perturbed into random noise, $\mathbf{x}_T \sim \mathcal{N}(0, \mathbf{I})$, where $T$ is the total number of diffusion steps. We follow the Variance Preserving stochastic differential equation (VP-SDE) [52, 20] where the forward process is modeled by the following SDE:

$$\mathrm{d}\mathbf{x}_t = -\frac{1}{2}\beta(t)\mathbf{x}_t\,\mathrm{d}t + \sqrt{\beta(t)}\mathrm{d}\mathbf{w}_t, \tag{1}$$

where $\beta(t)$ is a pre-defined noise schedule and $\mathbf{w}_t$ is a standard Wiener process. Then, the reverse of the forward process in Eq. (1) is also a diffusion process that is modeled by the following SDE [2]:

$$\mathrm{d}\mathbf{x}_t = \left[-\frac{1}{2}\beta(t)\mathbf{x}_t - \beta(t)\nabla_{\mathbf{x}_t}\log p_t(\mathbf{x}_t)\right]\mathrm{d}t + \sqrt{\beta(t)}\mathrm{d}\bar{\mathbf{w}}_t, \tag{2}$$

where $p_t$ is the probability density of $\mathbf{x}_t$ and $\bar{\mathbf{w}}_t$ is a standard Wiener process with backward time flows. The reverse process in Eq. (2) can be used as a generative model when the score function $\nabla_{\mathbf{x}}\log p_t(\mathbf{x})$ is known. To estimate the score function from given data, a neural network $\mathbf{s}_{\boldsymbol{\theta}}$ is trained to minimize the following objective [52]:

$$\mathcal{L}(\boldsymbol{\theta}) = \mathbb{E}_{t,\mathbf{x}_0}\lambda(t)\|\mathbf{s}_{\boldsymbol{\theta}}(\mathbf{x_t}, t) - \nabla_{\mathbf{x}_t}\log p_t(\mathbf{x}_t)\|_2^2, \tag{3}$$

where $t \sim U[0, T]$, $\mathbf{x}_0$ is sampled from the data distribution, and $\lambda : [0, T] \to \mathbb{R}^+$ is a positive weight function.

**Online guidance** Recent works [17, 9, 51] leverage a conditional diffusion process where the conditional probability $p_t(\mathbf{x}_t|\mathbf{c})$ is modeled without training on labeled pairs $(\mathbf{x}_t, \mathbf{c})$. To understand this approach, observe that for the conditional generation, the reverse SDE of Eq. (2) can be rewritten as follows:

$$\mathrm{d}\mathbf{x}_t = \left[ -\frac{1}{2}\beta(t)\mathbf{x}_t - \beta(t)\nabla_{\mathbf{x}_t} \log p_t(\mathbf{x}_t|\mathbf{c}) \right] \mathrm{d}t + \sqrt{\beta(t)}\mathrm{d}\bar{\mathbf{w}}_t. \tag{4}$$

From Bayes' rule, the conditional score $\nabla_{\mathbf{x}} \log p_t(\mathbf{x}_t|\mathbf{c})$ can be decomposed as follows:

$$\nabla_{\mathbf{x}_t} \log p_t(\mathbf{x}_t|\mathbf{c}) = \nabla_{\mathbf{x}_t} \log p_t(\mathbf{x}_t) + \nabla_{\mathbf{x}_t} \log p_t(\mathbf{c}|\mathbf{x}_t). \tag{5}$$

While unconditional score $\nabla_{\mathbf{x}_t} \log p_t(\mathbf{x}_t)$ can be approximated by the unconditional diffusion model $s_{\boldsymbol{\theta}}$ from Eq. (3), online guidance can estimate conditional score part $\nabla_{\mathbf{x}_t} \log p(\mathbf{c}|\mathbf{x}_t)$ without any training. Notably, DPS [9] approximates conditional score as follows:

$$\nabla_{\mathbf{x}_t} \log p(\mathbf{c}|\mathbf{x}_t) \approx \nabla_{\mathbf{x}_t} \log p(\mathbf{c}|\hat{\mathbf{x}}_0), \tag{6}$$

where $\hat{\mathbf{x}}_0$ is a point estimation of the final clean data using Tweedie's formula [13]:

$$\hat{\mathbf{x}}_0 = \frac{\mathbf{x}_t + (1 - \bar{\alpha}_t)\nabla_{\mathbf{x}_t} \log p(\mathbf{x}_t)}{\sqrt{\bar{\alpha}_t}}, \quad \bar{\alpha}_t = e^{-\frac{1}{2}\int_0^t \beta(s)ds}. \tag{7}$$

While DPS uses a point estimate of $\mathbf{x}_0$, Loss Guided Diffusion (LGD) [51] uses Bayesian assumption where $\mathbf{x}_0$ is a Monte Carlo estimation of $q(\mathbf{x}_0|\mathbf{x}_t)$, which is a normal distribution with mean $\hat{\mathbf{x}}_0$ and variance $\sigma_t^2$ which is a hyperparameter.

For a given property estimator $\mathcal{A}$ satisfying $\mathbf{c} = \mathcal{A}(\mathbf{x}_0)$, Eq. (6) can be written as follows:

$$\nabla_{\mathbf{x}_t} \log \mathbb{E}_{\mathbf{x}_0 \sim p(\mathbf{x}_0|\mathbf{x}_t)} p(\mathbf{c}|\hat{\mathbf{x}}_0) \approx \nabla_{\mathbf{x}_t} \log \left( \frac{1}{m} \sum_{i=1}^m \exp\left( -\mathcal{L}(\mathcal{A}(\mathbf{x}_0^i), \mathbf{c}) \right) \right) =: \mathbf{g}(\mathbf{x}_t, t), \tag{8}$$

where $\mathcal{L}$ is a differentiable loss function and $\mathbf{x}_0^i$ are independent variables sampled from $q(\mathbf{x}_0|\mathbf{x}_t)$. Using this approximation, we can replace $\nabla_{\mathbf{x}_t} \log p(\mathbf{c}|\mathbf{x}_t)$ in Eq. (5) and conditional generation process now becomes:

$$\mathrm{d}\mathbf{x}_t = \left[ -\frac{1}{2}\beta(t)\mathbf{x}_t - \beta(t) \left( \nabla_{\mathbf{x}_t} \log p_t(\mathbf{x}_t) + z\mathbf{g}(\mathbf{x}_t, t) \right) \right] \mathrm{d}t + \sqrt{\beta(t)}\mathrm{d}\bar{\mathbf{w}}_t, \tag{9}$$

where $z$ is the hyperparameter for controlling online guidance strength. In this work, we use $L_2$ loss for the differentiable loss $\mathcal{L}$ and set $m = 1$ following Song et al. [51].

**3D representations of molecules** In our framework, we follow EDM [23] in modeling the distribution of atomic coordinates $\mathbf{x}$ within a linear subspace $\mathcal{X}$, where the center of mass is constrained to zero. This allows for translation-invariant modeling of molecular geometries. Further details of the zero-center-of-mass subspace is provided in Appendix A.3.

## 4 Time-Aware Conditional Synthesis

In this section, we propose our framework, Time-Aware Conditional Synthesis (TACS). Section 4.1 presents the key component of TACS: the Time Correction Sampler, a novel sampling technique that leverages corrected time information during the generation process. Section 4.2 introduces the overall framework of TACS, which accurately integrates online guidance into the diffusion process using our sampling technique to generate stable and valid molecules that meet the target conditions.

### 4.1 Time Correction Sampler

Diffusion models are known to have an inherent bias, referred to as exposure bias, where the marginal distributions of the forward process do not match the learned marginal distributions of the backward process [40]. To mitigate exposure bias in the diffusion models, we propose the Time Correction Sampler (TCS), which consists of two parts: time prediction and time correction.

---

**Algorithm 1** Time-Aware Conditional Synthesis (TACS)

---

**Input:** Total number of diffusion timesteps $T$, online guidance strength $z$, target condition $\mathbf{c}$, diffusion model $\boldsymbol{\theta}$, time predictor $\phi$, time-clip window size $\Delta$.

1: $\mathbf{x}_T \sim \mathcal{N}(0, \mathbf{I}_d)$
2: **for** $t = T$ to $1$ **do**
3:     **if** Online guidance **then**
4:         $\mathbf{g}(\mathbf{x}_t, t) = -\nabla_{\mathbf{x}_t} \mathcal{L}(\mathcal{A}(\mathbf{x}_0), \mathbf{c})$                              $\triangleright$ Online guidance from Eq. (8)
5:         $\mathbf{x}'_t \leftarrow \mathbf{x}_t + z \cdot \mathbf{g}(\mathbf{x}_t, t)$
6:         $t_{\text{pred}} \leftarrow \arg\max(\boldsymbol{\phi}(\mathbf{x}'_t))$
7:         $t_{\text{pred}} \leftarrow \texttt{clip}(t_{\text{pred}}, \Delta)$
8:         $\hat{\mathbf{x}}'_0 \leftarrow \texttt{Tweedie}(\mathbf{x}'_t, t_{\text{pred}})$                                    $\triangleright$ from Eq. (11)
9:         $\mathbf{x}_{t-1} \leftarrow \texttt{forward}(\hat{\mathbf{x}}'_0, t-1)$                           $\triangleright$ from Eq. (1)
10:     **else**
11:         $\mathbf{x}_{t-1} \leftarrow \texttt{reverse}(\mathbf{x}_t, t)$             $\triangleright$ one reverse step by diffusion model
12:     **end if**
13: **end for**

---

**Time Predictor** During conditional generation process of diffusion model, a sample follows the reverse SDE as in Eq. (4). However, due to error accumulation during the generation process [33, 6], a sample at timestep $t$ of the reverse process of diffusion may not accurately reflect the true marginal distribution at timestep $t$. This discrepancy between forward and reverse process can especially increased when applying online guidance for every denoising steps and consequently lead to the generation of molecules with low stability and validity. We aim to mitigate this issue by correcting the time information based on the sample's current position.

To achieve this, we train a neural network, a time predictor, to estimate the proper timestep of a given noised data. Specifically, given a random data point $\mathbf{x}$ with unknown timestep, time predictor models how likely $\mathbf{x} \sim p_t$ for each timestep in $[0, T]$. For training, we parameterize a time predictor by equivariant graph neural network (EGNN) $\phi$. Then, cross-entropy loss between the one hot embedding of timestep vector and the logit vector for the model output is used as follows:

$$\mathcal{L}_{\text{tp}}(\phi) = -\mathbb{E}_{t, \mathbf{x}_0}\left[\log(\hat{\mathbf{p}}_\phi(\mathbf{x}_t)_t)\right], \tag{10}$$

where timestep $t$ is sampled from the uniform distribution $U[0, T]$, $\mathbf{x}_0$ is chosen from the data distribution, $\mathbf{x}_t$ is constructed from Eq. (1), and $\hat{\mathbf{p}}_\phi(\mathbf{x}_t)_t$ is the $t$-th component of the model output $\hat{\mathbf{p}}_\phi$ for a given input $\mathbf{x}_t$. We empirically evaluate the performance of the time predictor on the QM9 train and test datasets. Forward noise, corresponding to each true timestep, is added to the data, and the time predictor estimates the true timestep, with accuracy measured accordingly. As shown in 3b, the predictor struggles in the white noise region, but achieves near-perfect accuracy after timestep 400.

**Time correction** TCS works by first modifying Tweedie's formula (Eq. 7) using the estimated timestep from time predictor to utilize the information of the proper timestep estimated by time predictor during the denoising diffusion process.

Specifically, for a sample $\tilde{\mathbf{x}}_t$ at time $t$ during the reverse diffusion process, we use the corrected timestep $t_{\text{pred}}$ predicted by time predictor, instead of the current timestep $t$, to estimate the final sample $\hat{\mathbf{x}}_0$ as follows:

$$\texttt{Tweedie}(\tilde{\mathbf{x}}_t, t_{\text{pred}}) := f(\tilde{\mathbf{x}}_t, t_{\text{pred}}) = \frac{\tilde{\mathbf{x}}_t + (1 - \bar{\alpha}_{t_{\text{pred}}}) s_{\boldsymbol{\theta}}(\tilde{\mathbf{x}}_t, t_{\text{pred}})}{\sqrt{\bar{\alpha}_{t_{\text{pred}}}}}. \tag{11}$$

From the better prediction of the final sample $\hat{\mathbf{x}}_0$, we perturb it back to the next timestep $t-1$ using the forward process using Eq. (1).

Intuitively, TCS encourages the generated samples to adhere more closely to the proper data manifolds at each denoising step. The iterative correction of the time through the generative process ensures the final sample lie on the correct data distribution. Consequently, for 3D molecular generation, TCS helps synthesizing molecules with higher molecular stability and validity, both of which are crucial for generating realistic and useful molecules.

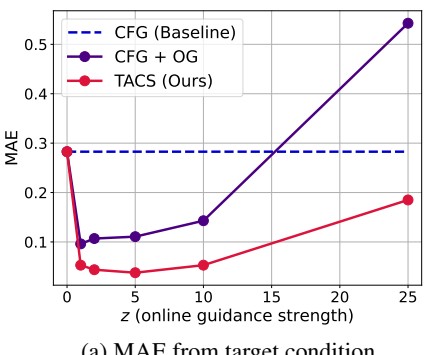 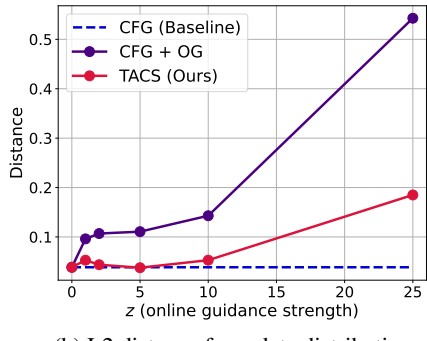

(a) MAE from target condition  (b) L2 distance from data distribution

Figure 2: Synthetic experiment on $H_3^+$ dataset. TACS is robust in generating samples that (a) match the desired condition and (b) stick to the original data distribution.

## 4.2 Unified guidance with time correction

Finally, we present TACS in Algorithm 1, which is a novel diffusion sampler for conditional generation that integrates TCS with online guidance. For each timestep of the reverse diffusion process (Eq. 4), we apply online guidance (Eq. 8) to guide the process toward satisfying the target condition. Subsequently, TCS is applied to correct the deviation from the proper marginal distribution induced by online guidance. This ensures the samples follow the correct marginal distribution during the generative process, as shown in Fig. 1. In Section 5, we experimentally validate that TACS is capable of generating valid and stable molecules satisfying the target condition.

**Time clipping**   During the generation, we empirically observe that the predicted timestep $t_{\text{pred}}$ often deviates significantly from the current time step $t$. Naively applying this information to TCS can be problematic as the dramatic changes in timestep may skip crucial steps, resulting in unstable or invalid molecules. To prevent large deviation of $t_{\text{pred}}$, we set a time window so that $t_{\text{pred}}$ remains in the time interval $[t - \Delta, t + \Delta]$, where $\Delta$ is a hyperparameter for the window size.

## 5 Experiments

In this section, we present comprehensive experiments to evaluate the performance of TACS and demonstrate its effectiveness in generating 3D molecular structures with specific properties while maintaining stability and validity. In Section 5.1, we present synthetic experiment with $H_3^+$ molecules, where the ground state energies are computed using the variational quantum eigensolver (VQE). In Section 5.2, we assess our method using QM9, a standard dataset in quantum chemistry that includes molecular properties and atom coordinates. We compare our approach against several state-of-the-art baselines and provide a detailed analysis of the results.

### 5.1 Synthetic experiment with $H_3^+$

**Quantum online guidance**   We present quantum online guidance as a modified version of online guidance where quantum machine learning algorithm for the property estimation of generating molecules. Specifically, we use VQE [53] when calculating the ground state energy of generated molecules. Contrary to prior works [23, 5], which train auxiliary classifiers to estimate each condition, this quantum computational chemistry-based approach can leverage exact calculation of the condition for a given estimate. This, in turn, is expected to generate molecules with accurate target ground state energies. A detailed explanation of this approach is provided in Appendix A.2.

**Setup**   We first construct synthetic $H_3^+$ as follows. For each molecule, a hydrogen atom is placed uniformly at random within the 3-D unit sphere. We then augment our sample by rotating each molecule randomly to satisfy the equivariance property [23]. Then the ground state energy of each molecule is measured with VQE in order to provide conditional labels. Finally, we train unconditional

Table 1: Conditional generation with target quantum properties on QM9. TACS generate samples with lowest MAE while maintaining similar level of stability and validity as other baselines. TCS can generate molecules with high stability and validity. $*$ notation is marked for the values for the best MAE within methods with molecule stability above 80%.

| | $C_v\left(\frac{\text{cal}}{\text{mol K}}\right)$ | | | $\mu$ (D) | | | $\alpha$ (Bohr$^3$) | | |
|---|---|---|---|---|---|---|---|---|---|
| Method | MAE | MS (%) | Valid (%) | MAE | MS (%) | Valid (%) | MAE | MS (%) | Valid (%) |
| U-bound | 6.879±0.015 | - | - | 1.613±0.003 | - | - | 8.98±0.020 | - | - |
| EDM | 1.072±0.005 | 81.0±0.6 | 90.4±0.2 | 1.118±0.006 | 80.8±0.2 | 91.2±0.3 | 2.77±0.050 | 79.6±0.3 | 89.9±0.2 |
| EEGSDE | 1.044±0.023 | 81.0±0.5 | 90.5±0.2 | 0.854±0.002 | 79.4±0.1 | 90.6±0.5 | 2.62±0.090 | 79.8±0.3 | 89.2±0.3 |
| OG | **0.184**±0.004 | 30.2±0.4 | 51.9±6.8 | 26.654±8.878 | 6.8±0.5 | 29.7±0.3 | **0.55**±0.019 | 21.2±0.2 | 48.7±0.3 |
| TCS(ours) | 0.904±0.011 | **90.3**±0.5 | **94.9**±0.1 | 0.971±0.005 | **92.7**±0.3 | **96.6**±0.2 | 1.85±0.006 | **87.5**±0.2 | **93.8**±0.1 |
| TACS(ours) | **0.659**$^*$±0.008 | 83.6±0.2 | 92.4±0.1 | **0.387**$^*$±0.006 | 83.3±0.3 | 91.3±0.3 | **1.44**$^*$±0.007 | 86.0±0.1 | 92.5±0.1 |
| L-bound | 0.040 | - | - | 0.043 | - | - | 0.090 | - | - |

| | $\Delta\epsilon$ (meV) | | | $\epsilon_{\text{HOMO}}$ (meV) | | | $\epsilon_{\text{LUMO}}$ (meV) | | |
|---|---|---|---|---|---|---|---|---|---|
| Method | MAE | MS (%) | Valid (%) | MAE | MS (%) | Valid (%) | MAE | MS (%) | Valid (%) |
| U-bound | 1464±4 | - | - | 645±41 | - | - | 1457±5 | - | - |
| EDM | 673±7 | 81.8±0.5 | 90.9±0.3 | 372±1 | 79.6±0.1 | 91.6±0.2 | 602±4 | 81.0±0.2 | 91.4±0.5 |
| EEGSDE | 539±5 | 80.1±0.4 | 90.5±0.3 | 300±5 | 78.7±0.7 | 91.2±0.2 | 494±9 | 81.4±0.6 | 91.1±0.2 |
| OG | **95.2**±4 | 31.3±0.7 | 61.9±3.4 | **233**±8 | 11.5±0.4 | 40.8±2.6 | **170**±1 | 21.1±0.5 | 42.3±0.9 |
| TCS(ours) | 594±4 | **91.9**±0.4 | **96.0**±0.2 | 338±5 | **92.7**±0.2 | **96.6**±0.2 | 493±9 | **91.7**±3.9 | **96.2**±0.4 |
| TACS(ours) | **332**$^*$±3 | 88.8±0.6 | 93.9±0.3 | **168**$^*$±2 | 87.3±0.7 | 93.0±0.2 | **289**$^*$±4 | 82.7±0.7 | 91.3±0.2 |
| L-bound | 65 | - | - | 39 | - | - | 36 | - | - |

diffusion model and CFG-based conditional diffusion model with the constructed data. For evaluation, we measure the ground state energy and calculate MAE (Mean Absolute Error) with the target energy. Also, we measure the average L2 distance between position of each atom and its projection to the unit sphere.

**Results**  The results in Figure 2 indicate that while online guidance correctly guides the samples to the target condition, it can lead to the samples deviated from the original data distribution. In contrast, molecules sampled from TACS can satisfy both low MAE and low L2 distance. Further details and additional discussions are provided in Appendix B.1.

### 5.2 Conditional generation for target quantum chemical properties

**Dataset**  We evaluate our method on QM9 dataset [45], which contains about 134k molecules with up to 9 heavy atoms of (C, N, O, F), each labeled with 12 quantum chemical properties. Following previous works [1, 23], we test on 6 types of quantum chemical properties and split the dataset into 100k/18k/13k molecules for training, validation, and test. The training set is further divided into two disjoint subsets of 50k molecules each: $D_a$ for property predictor training and $D_b$ for generative model training. Further details are provided in Appendix B.2.

**Evaluation**  To evaluate how generate samples meet the desired target condition, a property prediction model $\phi_p$ [47] is trained on $D_a$. Then, MAE for $K$ number of samples is calculated as $\frac{1}{K}\sum_{i=1}^{K}|\phi_p(\mathbf{x}^{(i)}) - c^{(i)}|$, where $\mathbf{x}^{(i)}$ represents $i$-th generated molecule and $c^{(i)}$ is corresponding target quantum chemical properties. Molecular stability (MS) and validity (Valid) [23] are used to measure how generated samples satisfy basic chemical properties. Details of the evaluation metrics are provided in Appendix B.2.

**Baselines**  We use Equivariant Diffusion Models (EDM) [23] and Equivariant Energy Guided SDE (EEGSDE) [5] for the baselines. Following [23], we put additional baselines including "Naive Upper-Bound" (randomly shuffled property labels), "#Atoms" (properties predicted by atom count), and "L-Bound" (lower bound on MAE using a separate prediction model).

**Results**  Table 1 shows the result of conditional generation of TACS and TCS with baselines. We generate $K=10^4$ samples for the evaluation in each experiment and the average values and standard deviations are reported across 5 runs. For all of the quantum chemical properties, TACS achieves lower MAE while maintaining comparable or higher molecular stability (MS) and validity compared to other baselines. Notably, when comparing with the baseline methods that maintains the MS above 80%, the MAE of TACS is significantly lower than the baselines. The result demonstrates the

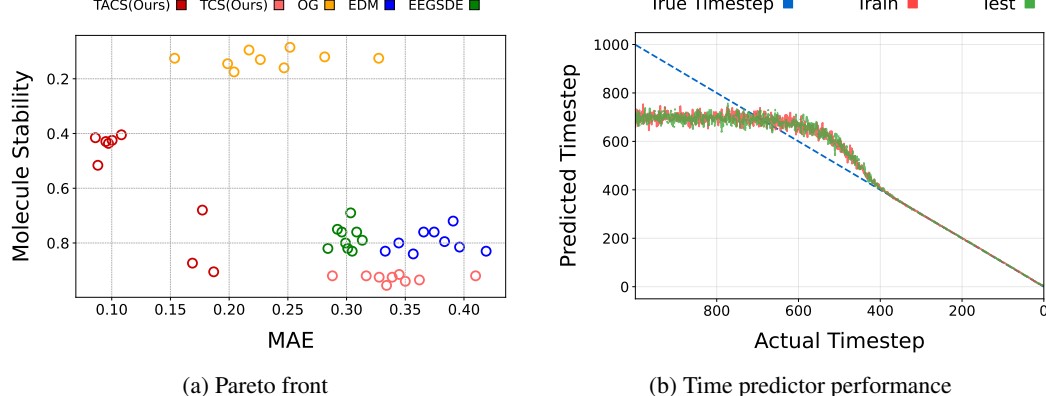

(a) Pareto front                                    (b) Time predictor performance

Figure 3: (a) Comparative analysis of molecule stability and MAE across five distinct methods, highlighting the enhanced stability of our approach at comparable MAE levels. (b) Performance of the time predictor for train and test set on QM9.

Table 2: Quantitative results showing similarity and stability of generated molecules compared to target structures.

| Method | QM9 | |
|---|---|---|
| | Similarity ↑ | MS (%) |
| cG-SchNet | $0.499\pm0.002$ | - |
| Conditional EDM | $0.671\pm0.004$ | - |
| TCS (Ours) | $\mathbf{0.792\pm0.077}$ | 90.42 |
| TACS (Ours) ($z = 0.01$) | $0.694\pm0.001$ | $\mathbf{90.45}$ |
| TACS (Ours) ($z = 0.05$) | $0.695\pm0.003$ | 90.02 |
| TACS (Ours) ($z = 0.1$) | $0.713\pm0.087$ | 90.28 |
| EEGSDE ($s = 0.1$) | $0.547\pm0.002$ | 74.07 |
| EEGSDE ($s = 0.5$) | $0.600\pm0.002$ | 74.67 |
| EEGSDE ($s = 1.0$) | $0.540\pm0.029$ | 90.44 |

Table 3: Unconditional generation using Geom-Drug dataset.

| Method | Geom-Drug | |
|---|---|---|
| | AS (%) ↑ | Valid (%) |
| Data | 86.5 | 99.9 |
| ENF | - | - |
| G-Schnet | - | - |
| GDM | 75.0 | 90.8 |
| GDM-AUG | 77.7 | 91.8 |
| EDM | 81.3 | 92.6 |
| EDM-Bridge | 82.4 | 92.8 |
| TCS(Ours) | $\mathbf{89.6}$ | $\mathbf{97.6}$ |

effectiveness of TACS in balancing the objectives of generating molecules with desired properties and ensuring their structural validity.

TCS achieves high validity and atom stability and molecular stability, surpassing the unconditional generation performance of the baselines, but with a higher MAE compared to TACS. This highlights the importance of online guidance for precise property targeting. However, applying only online guidance yields samples with low MAE but suffers from reduced validity and stability, occasionally failing to generate valid molecules due to numerical instability. This shows the ability of TACS which places the samples on the correct data manifold at each denoising step, even if they deviate from the true time-manifold due to the online guidance. Finally, we put the results of different methods with 9 different runs for each method and plot the MAE and MS in Figure 3a. The result clearly shows that TACS and TCS reaches closer to the Pareto front of satisfying molecular stability and the target condition together.

## 5.3 Target structure generation

We conduct experiments on target structure generation with QM9 as in [5]. For evaluation, we report Tanimoto similarity score [16] which captures similarity of molecular structures by molecular fingerprint and molecular stability to check whether basic properties of molecules are satisfied during the conditional generation process. We put additional details of the experiment in Appendix B.3.2.

**Results** Table 2 shows that TACS significantly outperforms baseline methods both in Tanimoto similarity and molecular stability. Interestingly, performance of TACS is robust in the online guidance strength $z$. This demonstrates TACS's ability to generalize on different tasks.

Table 4: Comparison between performance of TACS when using argmax function and expectation for correcting timesteps using time predictor.

| Method | $C_v \left(\frac{\text{cal}}{\text{mol K}}\right)$ | | $\mu$ (D) | | $\alpha$ (Bohr$^3$) | | $\Delta\epsilon$ (meV) | | $\epsilon_{\text{HOMO}}$ (meV) | | $\epsilon_{\text{LUMO}}$ (meV) | |
|---|---|---|---|---|---|---|---|---|---|---|---|---|
| | MAE | MS (%) | MAE | MS (%) | MAE | MS (%) | MAE | MS (%) | MAE | MS (%) | MAE | MS (%) |
| Argmax | 0.659 | 83.3 | 0.387 | 83.3 | 1.44 | 86.0 | 332 | 88.8 | 168 | 87.3 | 289 | 82.7 |
| Expectation | 0.703 | 84.6 | 0.451 | 90.2 | 1.56 | 87.3 | 351 | 90.7 | 182 | 89.8 | 334 | 90.1 |

## 5.4 Unconditional generation on Geom-Drug dataset

To verify the scalability of time correction sampler, we use Geom-Drug [4] as our dataset. Geom-Drug consists of much larger and complicate molecules compared to the QM9. For fair comparisons, we follow [23, 5] to split the dataset for training, validation and test set include 554k, 70k, and 70k samples respectively. We test the performance of TCS on unconditional generation of 3D molecules when using Geom-Drug. For evaluation, we use atom stability (AS) and validity of generated samples. The result in Table 3 shows that samples generated by TCS satisfy the highest atom stability and the validity with high margin compared to the baselines. Details of the experiments are in Appendix B.3.1.

## 5.5 Ablation Studies

**Time prediction function**  We investigate how the the design of time prediction function affects the performance of TACS. Specifically, for line 6 in Alg. 1, rather than using argmax function to obtain corrected timestep $t_{\text{pred}}$, we choose to use expectation value by $t_{\text{pred}} = \mathbb{E}[\phi(x)]$. Table 4 shows the comparison between two methods in six different types of quantum chemical properties. Expectation based time prediction results in molecules with higher MAE and higher molecular stability.

**Online guidance strength** $z$  To analyze the effect of online guidance strength $z$ in Eq. (9), we measure MAE, molecular stability, and validity of samples generated by TACS for different $z$ values. Table 5 shows the result with target condition on $\epsilon_{\text{LUMO}}$ values. One can observe while trade-off occurs for different $z$ values, performance of TACS is robust in varying $z$. Interestingly, our experiment shows that there exists an optimal value of $z$ which generates samples with the lowest MAE that is even compara-

Table 5: Ablation study on the OG strength $z$ with target property $\epsilon_{\text{LUMO}}$.

| Method | MAE | MS (%) | Valid (%) |
|---|---|---|---|
| TCS ($z = 0$) | 493 | **91.7** | **96.2** |
| OG | **170** | 21.1 | 42.3 |
| $z = 1.5$ | 311 | 80.3 | 90.7 |
| $z = 1.0$ | 236 | 74.9 | 86.3 |
| $z = 0.5$ | 288 | 82.7 | 91.3 |

ble to applying online guidance without time correction (OG). As expected, using $z = 0$ (TCS) generates molecules with the highest MS and validity but with the highest MAE.

**Time window length** $\Delta$  We measure how the performance of TACS varies with time window length $\Delta$. Table 6 shows the MAE, molecular stability values for different window sizes when the target property is $\alpha$. The result shows that the performance of TACS is robust when using moderate window size but decreases when window size becomes larger than certain point. We set $\Delta = 10$ for other experiments.

In Appendix C, we provide the results of further ablation studies including the effect of mc sampling and effective diffusion steps for TACS. Overall, the results show that our method is robust in the choice of hyperparameters and generalizable to different datasets and tasks.

Table 6: Results of TACS on target property $\alpha$ when varying the time window size.

| Window size ($\Delta$) | MAE | MS (%) |
|---|---|---|
| 2 | 1.481 | **87.25** |
| 4 | 1.460 | 86.52 |
| 6 | 1.460 | 85.73 |
| 8 | 1.448 | 86.84 |
| 10 | **1.441** | 86.08 |
| 12 | 1.442 | 85.17 |
| 14 | 1.459 | 82.76 |
| 16 | 1.530 | 79.10 |

## 6 Discussion

**Exploiting quantum chemistry**  In Section 5.1, we demonstrate that quantum computing-based guidance can serve as an accurate property predictor for the online guidance. Currently, in the absence of a non-noisy quantum computer, scaling up this exact guidance to the QM9 dataset is close to impossible due to compounding noise [43, 10]. However, future fault-tolerant quantum technology is

expected to provide quantum advantage in calculating chemical properties. This can be incorporated into our algorithm when using online guidance and therefore, further improvements of our TACS are on the horizon.

**Connection to other fields**   Recent works point out the exposure bias exists for diffusion models [40], where there is a mismatch between forward and reverse process. Our experiments in 5.2 indicate that Time Correction Sampler can provide a solution to the exposure bias problem during the sampling process in diffusion models. Moreover, since time predictor can gauge this mismatch during inference, one might leverage this information for future works.

Another direction is applying our algorithm to matching [56, 27, 39] frameworks. Contrary to diffusion models, these matching models can start with arbitrary distributions and directly learn vector fields. We expect time predictor to be also effective with these types of algorithms. Investigating the connection between our algorithm and matching models will be an interesting future direction.

# 7   Conclusion

In this work, we introduce Time-Aware Conditional Synthesis (TACS), a novel approach for conditional 3D molecular generation using diffusion models. Our algorithm leverages a Time Correction Sampler (TCS) in combination with online guidance to ensure that generated samples remain on the correct data manifold during the reverse diffusion process. Our experimental results clearly demonstrate the advantage of our algorithm, as it can generate molecules that are close to the target conditions while also being stable and valid. This can be seen as a significant step towards precise and reliable molecular generation.

Despite multiple advantages, several open questions remain. For example, how can we more efficiently use the Time Correction Sampler, or more generally, whether this method improves performance in other domains such as in image generation. We expect that our work will open various opportunities across different domains, such as quantum chemistry and diffusion models.

**Limitation**   Although we demonstrate the effectiveness of our algorithm on multiple datasets and tasks, we use a trained neural network to estimate chemical properties of each molecule for main experiments. Using exact computational chemistry-based methods might improve our algorithm.

**Societal impacts**   We believe that our framework can assist in drug discovery, which requires synthesizing stable and valid molecules that satisfy target conditions. However, our work could unfortunately be misused to generate toxic or harmful substances.

# Acknowledgments

This work was supported by Institute for Information & communications Technology Planning & Evaluation (IITP) grant funded by the Korea government (MSIT) (No. RS-2019-II190075, Artificial Intelligence Graduate School Program (KAIST); No. RS-2024-00457882, AI Research Hub Project), the National Research Foundation of Korea NRF grant funded by the Korean government (MSIT) (No. RS-2019-NR040050 Stochastic Analysis and Application Research Center (SAARC)), and LG Electronics.

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

# A  Method details

## A.1  Time Predictor

For model architecture, we use EGNN [47] with $L = 7$ layers, each with hidden dimension $h_f = 192$. We use the same split of training / test dataset of QM9.

**Time predictor training**  For completeness, we restate the training of the time predictor (Eq. 10) here. The time predictor is trained for minimizing the cross-entropy loss between the predicted logits $\hat{\mathbf{p}}$ and one-hot vector of the forward timestep $t$:

$$\mathcal{L}_{\text{time-predictor}}(\boldsymbol{\phi}) = -\mathbb{E}_{t, \mathbf{x}_0} \left[ \log \left( \hat{\mathbf{p}}_{\boldsymbol{\phi}}(\mathbf{x}_t)_t \right) \right], \tag{12}$$

where $t$ is uniformly drawn from the interval $[0, T]$.

For conditional diffusion model, we train separate time predictor for each of the target property by concatenating conditional information $c$ to the input $\mathbf{x}_t$.

The rationale behind using cross-entropy loss rather than simple regression loss is because $p(t|\mathbf{x})$ can not be estimated from the point estimate if there exists intersection between support of marginal distributions $p_t$ and $p_s$ which is from different timesteps $s$ and $t$, respectively. In other words, this implies there exists $\mathbf{x} \in \mathcal{R}^d$ such that $p_t(\mathbf{x}) > 0$ and $p_s(\mathbf{x}) > 0$ simultaneously holds.

Finally, we train the time predictor within 24 hours with 4 NVIDIA A6000 GPUs.

## A.2  Quantum online guidance

**Quantum Machine Learning**  Quantum computing is expected to become a powerful computational tool in the future [14]. While at its early stage, various of quantum machine learning algorithms are proved to have advantage over classical methods [3]. In computational chemistry, these advantages are indeed expected to have huge potential since classically intractable computations like finding ground state for big molecules are expected to become feasible with exponential speed-ups [48] of quantum machines.

**Variational Quantum Eigensolver**  Variational Quantum Eigensolver (VQE) [41] is a near term quantum machine learning algorithm which leverages variational principle to obtain the lowest energy of a molecule with given Hamiltonian. For the given Hamiltonian $\hat{H}$, trial wave function $|\psi(\boldsymbol{\theta})\rangle$ which is parameterized by a quantum circuit ($\boldsymbol{\theta}$) is prepared to obtain ground state energy $E_0$ with following inequality:

$$E_0 \leq \frac{\langle \psi(\boldsymbol{\theta}) | \hat{H} | \psi(\boldsymbol{\theta}) \rangle}{\langle \psi(\boldsymbol{\theta}) | \psi(\boldsymbol{\theta}) \rangle}. \tag{13}$$

Specifically, parameters $\boldsymbol{\theta}$ in quantum circuit is iteratively optimized to minimize the following objective:

$$E(\boldsymbol{\theta}) = \frac{\langle \psi(\boldsymbol{\theta}) | \hat{H} | \psi(\boldsymbol{\theta}) \rangle}{\langle \psi(\boldsymbol{\theta}) | \psi(\boldsymbol{\theta}) \rangle}. \tag{14}$$

When quantum circuit $\boldsymbol{\theta}$ is expressive enough, one can see that $|\psi(\boldsymbol{\theta}_\star)\rangle$, where $\boldsymbol{\theta}_\star$ is a minimizer of Eq. (14), gives the ground state of the given system. For more comprehensive review with its potential advantages, one may refer to [53].

**Quantum online guidance**  Quantum online guidance is a type of online guidance algorithm where we use VQE-based algorithm to calculate exact values of quantum chemical properties instead of using classifier which is usually a neural network.

In each denoising step of diffusion model, we first apply Tweedie's formula (Eq. 7) to estimate clean molecule $\hat{\mathbf{x}}_0$. Then we use VQE to calculate ground state energy of the estimated molecule by iteratively updating $\boldsymbol{\theta}$ for $E(\hat{\mathbf{x}}_0, \boldsymbol{\theta}) = \langle \psi(\boldsymbol{\theta}) | H(\hat{\mathbf{x}}_0) | \psi(\boldsymbol{\theta}) \rangle$. After obtaining $\boldsymbol{\theta}_\star(\hat{\mathbf{x}}_0)$ which minimizes $E(\hat{\mathbf{x}}_0, \boldsymbol{\theta})$, we obtain target property value $E_0(\hat{\mathbf{x}}_0)$. To obtain gradient in Eq. (8), we use zeroth-order method [35] with respect to the position of atoms to obtain gradient as follows:

$$\nabla_{\mathbf{x}_t} \log \mathbb{E}_{\mathbf{x}_0 \sim p(\mathbf{x}_0 | \mathbf{x}_t)} p(\mathbf{c} | \hat{\mathbf{x}}_0) \approx -\nabla_{\mathbf{x}_t} E_0(\hat{\mathbf{x}}_0^i) \approx -\sum_{i=1}^{k} \frac{E_0(\hat{\mathbf{x}}_0^i + \boldsymbol{h}_i) - E_0(\hat{\mathbf{x}}_0^i - \boldsymbol{h}_i)}{2\boldsymbol{h}_i}, \tag{15}$$

where $k$ is a hyperparameter for multipoint estimate when using the zeroth-order optimization.

### A.3 Properties of the Zero-Center-of-Mass Subspace

Let $\mathcal{X} = \{\mathbf{x} \in \mathbb{R}^{M \times 3} : \frac{1}{M} \sum_{i=1}^{M} \mathbf{x}_i = \mathbf{0}\}$ be the subspace of $\mathbb{R}^{M \times 3}$ where the center of mass is zero. Here we discuss some properties of $\mathcal{X}$ that are used in the TACS framework. First, note that $\mathcal{X}$ is a linear subspace of $\mathbb{R}^{M \times 3}$ with dimension $(M-1) \times 3$. There exists an isometric isomorphism $\phi : \mathbb{R}^{(M-1) \times 3} \to \mathcal{X}$, i.e., a linear bijective map that preserves distances: $\|\phi(\hat{\mathbf{x}})\| = \|\hat{\mathbf{x}}\|$ for all $\hat{\mathbf{x}} \in \mathbb{R}^{(M-1) \times 3}$. Intuitively, $\phi$ allows us to map between the lower-dimensional space $\mathbb{R}^{(M-1) \times 3}$ and the constrained subspace $\mathcal{X}$ without distortion. If $\mathbf{x} \in \mathcal{X}$ is a random variable with probability density $q(\mathbf{x})$, then the corresponding density of $\hat{\mathbf{x}} = \phi^{-1}(\mathbf{x})$ in $\mathbb{R}^{(M-1) \times 3}$ is given by $\hat{q}(\hat{\mathbf{x}}) = q(\phi(\hat{\mathbf{x}}))$. Similarly, a conditional density $q(\mathbf{x}|\mathbf{y})$ on $\mathcal{X}$ can be written as $\hat{q}(\hat{\mathbf{x}}|\hat{\mathbf{y}}) = q(\phi(\hat{\mathbf{x}})|\phi(\hat{\mathbf{y}}))$. In practice, computations involving probability densities on $\mathcal{X}$ can be performed in $\mathbb{R}^{(M-1) \times 3}$ and mapped back to $\mathcal{X}$ using $\phi$ as needed. This allows TACS to efficiently learn and sample from distributions on molecular geometries while preserving translation invariance. For further mathematical details on subspaces defined by center-of-mass constraints, we refer the reader to [29] and [47].

### A.4 Classfier-free guidance

For conditional generation, we need conditional score $\nabla_{\mathbf{x}} \log p_t(\mathbf{x}_t|\mathbf{c})$ where $\mathbf{c}$ is our target condition. Classifier-free guidance (CFG) [19] replaces conditional score with combination of unconditional score and conditional score. Here, diffusion model is trained with combination of unlabeled sample $(\mathbf{x}_0, \emptyset)$ and labeled samples $(\mathbf{x}_0, \mathbf{c})$. The reverse diffusion process (Eq. 2) in CFG changes as follows:

$$\mathrm{d}\mathbf{x}_t = \left[ -\frac{1}{2}\beta(t)\mathbf{x}_t - \beta(t)\left(-w\nabla_{\mathbf{x}_t} \log p_t(\mathbf{x}_t) + (1+w)\nabla_{\mathbf{x}_t} \log p_t(\mathbf{x}_t|\mathbf{c})\right) \right] \mathrm{d}t + \sqrt{\beta(t)}\,\mathrm{d}\bar{\mathbf{w}}_t \quad (16)$$

Here, $w$ is a conditional weight which controls the strength of conditional guidance. When $w = -1$, the process converges to the unconditional generation and with larger $w$, one can put more weights on conditional score. In the experiments, we use $w = 0$ following [23, 5].

## B Experimental details

### B.1 Synthetic experiment with $H_3^+$

Here, we provide experimental results on our synthetic experiment. Our results show that TACS is robust through different online guidance strength z and outperforms naive mixture of CFG and OG.

Table 7: MAE with target condition

| Z | CFG+OG | TACS |
|---|--------|------|
| 0 | 0.4823 | 0.0817 |
| 1 | 0.0961 | 0.0530 |
| 2 | 0.1069 | 0.0438 |
| 5 | 0.1107 | 0.0377 |
| 10 | 0.1430 | 0.0529 |
| 25 | 0.5427 | 0.3089 |

Table 8: L2 distance from data distribution

| Z | CFG+OG | TACS |
|---|--------|------|
| 0 | 0.0386 | 0.0294 |
| 1 | 0.0653 | 0.0305 |
| 2 | 0.1886 | 0.0339 |
| 5 | 0.6553 | 0.0383 |
| 10 | 3.5112 | 0.1820 |
| 25 | 18.8983 | 8.2466 |

**Geometric Optimization** It is known that $H_3^+$ molecule has a ground state energy around $-1.34$Ha. Assuming we don't have any prior knowledge of this information, we try to generate molecules with conditioning on the target ground state energy $-2.0$Ha. Applying TACS proves to be strong in this case also, which implies that our method can robustly guide the molecule without destroying distances.

## B.2 Experiments on QM9

**Dataset details** The QM9 dataset [45] is a widely-used benchmark in computational chemistry and machine learning. It contains 134k stable small organic molecules with up to 9 heavy atoms (C, O, N, F) and up to 29 atoms including hydrogen. This size constraint allows the molecules to be exhaustively enumerated and have their quantum properties calculated accurately using density functional theory (DFT). Each molecule in QM9 is specified by its Cartesian coordinates (in Angstroms) of all atoms at equilibrium geometry, along with 12 associated properties calculated from quantum mechanical simulations.

**QM9 molecular properties** In Table 9, we describe six quantum chemical properties that are used for conditional generation in our experiments.

Table 9: Quantum chemical properties

| Property | Description |
|---|---|
| Polarizability ($\alpha$) | Measure of a molecule's ability to form instantaneous dipoles in an external electric field. |
| HOMO-LUMO gap ($\Delta\epsilon$) | Energy gap between the HOMO and LUMO orbitals, indicating electronic excitation energies and chemical reactivity. |
| HOMO energy ($\epsilon_{\text{HOMO}}$) | Energy of the highest occupied molecular orbital, related to ionization potential and donor reactivity. |
| LUMO energy ($\epsilon_{\text{LUMO}}$) | Energy of the lowest unoccupied molecular orbital, related to electron affinity and acceptor reactivity. |
| Dipole moment ($\mu$) | Measure of the separation of positive and negative charges in a molecule, indicating polarity. |
| Heat capacity ($C_v$) | Measure of how much the temperature of a molecule changes when it absorbs or releases heat. |

**Performance metrics** To evaluate the quality of generated molecules, Table 10 shows descriptions of the metrics that are used in our experiments to check whether generated samples satisfy basic molecular properties.

Table 10: Performance metrics

| Property | Description |
|---|---|
| Validity (Valid) | Proportion of generated molecules that are chemically valid, as determined by RDKit. A molecule is valid if RDKit can parse it without encountering invalid valences. |
| Atom Stability (AS) | Percentage of atoms within generated molecules that possess correct valencies. Bond types are predicted based on atom distances and types, and validated against thresholds. An atom is stable if its total bond count matches the expected valency for its atomic number. |
| Molecule Stability (MS) | Percentage of generated molecules where all constituent atoms are stable. |

**Baselines** We compare our method against Equivariant Diffusion Models (EDM) [23] which learn a rotationally equivariant denoising process for property-conditioned generation and Equivariant Energy Guided SDE (EEGSDE) [5], which guides generation with a learned time-dependent energy function.

**Additional baselines** Two baselines are employed following the approach outlined by [23]. To measure "Naive (U-Bound)", we disrupt any inherent correlation between molecules and properties by shuffling the property labels in $\mathcal{D}_b$ and evaluating $\phi_c$ on the modified dataset. "L-Bound" is a lower bound estimation of the predictive capability. This value is obtained by assessing the loss of $\phi_c$ on $\mathcal{D}_b$, providing a reference point for the minimum achievable performance. If a proposed model, denoted

as, surpasses the performance of Naive (U-Bound), it indicates successful incorporation of conditional property information into generated molecules. Similarly, outperforming the L-Bound demonstrates the model's capacity to incorporate structural features beyond atom count and capture the intricacies of molecular properties. These baselines establish upper and lower bounds for evaluating mean absolute error (MAE) metrics in conditional generation tasks.

**Diffusion model training**  For a fair comparison with EDM and EEGSDE [23, 5], we adopt their training settings, using model checkpoints provided in the EEGSDE code: `https://github.com/gracezhao1997/EEGSDE`. The diffusion model is trained for 2000 epochs with a batch size of 64, learning rate of 0.0001, Adam optimizer, and an exponential moving average (EMA) with a decay rate of 0.9999. During evaluation, we generate molecules by first sampling the number of atoms $M \sim p(M)$ and the property value $\mathbf{c} \sim p(\mathbf{c}|M)$. Here $p(M)$ is the distribution of molecule sizes in the training data, and $p(\mathbf{c}|M)$ is the conditional distribution of the property given the molecule size. Then we generate a molecule conditioned on $M$ and $\mathbf{c}$ using the learned reverse process.

**Hyperparameters for TACS**  We analyze different hyperparameter settings for all six quantum chemical properties ($\alpha$, $\Delta\epsilon$, $\epsilon_{\text{HOMO}}$, $\epsilon_{\text{LUMO}}$, $\mu$, $C_v$) for the result in the Table 1. We vary the TCS starting timestep $t_{\text{TCS}} \in \{200, 400, 600, 800\}$, online guidance starting timestep $t_{\text{OG}} \in \{200, 400, 600, 800\}$, online guidance ending timestep $\tilde{t}_{\text{OG}} \in \{10, 20, 30\}$, gradient clipping threshold $\kappa \in \{\infty, 1, 0.1\}$ for Eq. (8), and guidance strength $z \in \{1.5, 1.0, 0.5\}$ in Eq. (9). Except for $\epsilon_{\text{LUMO}}$ (optimal with $z = 0.5$, $\tilde{t}_{\text{OG}} = 0$), we find $t_{\text{TCS}} = 600$, $t_{\text{OG}} = 600$, $\tilde{t}_{\text{OG}} = 20$, $z = 1$, and $\kappa = 1$ consistently achieve low MAE with molecular stability above 80%.

## B.3 Additional Experiment Details

### B.3.1 Experiments on Geom-Drug

**Dataset Details**  Geom-Drug dataset [4] contains approximately 450K molecules with up to 181 atoms and an average of 44.4 atoms per molecule. Following [5], we split the dataset into training/validation/test sets of 554k/70k/70k samples respectively.

**Performance Metrics**  For each experiment, we generate 10,000 molecular samples for evaluation. As in QM9 experiments, we measure atom stability (AS) and validity (Valid) using RdKit [32].

**Time Predictor Training**  For Geom-Drug, we employ TCS with 4 EGNN layers and 256 hidden features. The time predictor $\phi$ is trained unconditionally with the same EGNN architecture as QM9 for 10 epochs. During generation, TCS starts from timestep 600 and utilizes a window size of 10 for time correction.

### B.3.2 Experiments on Target Structure Generation

**Training Details**  For molecular fingerprint-based structure generation, we follow the evaluation protocol in [5]. The diffusion model architecture remains consistent with our QM9 experiments, using EGNN with 256 hidden features and 4 layers, trained for 10 epochs.

**Performance Metrics**  For evaluation, we use Tanimoto similarity score [16] which measures the structural similarity between generated molecules and target structures through molecular fingerprint comparison. Specifically, let $S_g$ and $S_t$ be the sets of bits that are set to 1 in the fingerprints of generated and target molecules respectively. The Tanimoto similarity is defined as $|S_g \cap S_t|/|S_g \cup S_t|$, where $|\cdot|$ denotes the number of elements in a set. We evaluate the similarity on 10,000 generated samples.

**Baselines**  For both experiments, we directly compare with baseline results reported in [22]. For Geom-Drug unconditional generation, these include ENF [47], G-Schnet [15], GDM variants [57], EDM [23], and EDM-Bridge [56]. For target structure generation, we compare against G-SchNet [15], GDM, GDM-AUG [57], Conditional EDM [23], and EEGSDE [5] with various guidance scales.

# C Additional experiments

## C.1 When to apply TCS and OG

We ablate when to start the time-corrected sampling (TCS) and online guidance (OG) during the reverse diffusion process. The notation $t_{\text{TCS}}$ and $t_{\text{OG}}$ indicates we apply TCS after timestep $t_{\text{TCS}}$ and OG after the timestep $t_{\text{OG}}$ in the reverse diffusion process. We report results for property $\epsilon_{\text{LUMO}}$ in Table 11. The result shows that applying TACS from early steps ($t = 600, 800$) generates samples best satisfying the target condition but with less molecular stability and validity. In contrast, when we start applying TACS after later step $t = 400$, generated samples have higher MAE, MS, and Validity. Interstingly, if we apply TACS only in the later part (after $t = 200$), MAE, MS, and validity decreases again. We leave further investigation on this phenomenon and explanation for future works.

In our experiments, we use $t_{\text{TCS}} = t_{\text{OG}} = 600$ as our default setting.

Table 11: Ablation study on when to start time-corrected sampling (TCS) and online guidance (OG) during the reverse diffusion process for the target property $\epsilon_{\text{LUMO}}$. We use $z = 1$ for the experiment. The best value in each column is bolded.

| $[t_{\text{TCS}}, t_{\text{OG}}]$ | MAE | MS (%) | Valid (%) |
|---|---|---|---|
| $[800, 800]$ | **236** | 74.9 | 86.3 |
| $[600, 600]$ | **236** | 74.9 | 86.2 |
| $[400, 400]$ | 360 | **86.6** | **93.3** |
| $[200, 200]$ | 248 | 72.1 | 84.5 |

## C.2 Number of MC samples

Additional experiments are conducted on the effect of number of MC samples $m$ in Eq. (8). LGD [51] estimates $\mathbf{x}_0$ assuming $q(\mathbf{x}_0|\mathbf{x}_t)$ is a normal distribution with mean $\hat{\mathbf{x}}_0$ (Eq. 7) and variance $\sigma^2$ which is a hyperparameter.

First, we investigate the effect of varying the variance $\sigma$ in MC sampling. Table 12 shows that the result is robust in the small values of $\sigma$ but when $\sigma$ is larger than some point, quality of generated samples decreases (higher MAE and lower MS).

Next, we test how the performance of TACS is affected by number of MC samples $m$. Table 13 shows that performance of TACS is robust in number of MC samples but we did not observe any performance increase with the number of MC samples as in [18].

Table 12: MC effect with varying $\sigma$. 5 MC samples are used and the target property is $\alpha$.

| $\sigma$ | MAE | MS (%) |
|---|---|---|
| 0.0001 | 1.506 | 86.06 |
| 0.0005 | 1.390 | 84.33 |
| 0.001 | 1.501 | 86.92 |
| 0.005 | 1.395 | 86.35 |
| 0.01 | 1.464 | 85.10 |
| 0.05 | 1.801 | 85.04 |
| 0.1 | 2.186 | 82.69 |
| 0.3 | 2.936 | 75.67 |

Table 13: Varying number of MC samples with $\sigma = 0.005$. Target property is $\alpha$.

| # Samples | MAE | MS (%) |
|---|---|---|
| 1 | 1.440 | 86.10 |
| 5 | 1.395 | 86.35 |
| 10 | 1.505 | 82.21 |
| 15 | 1.545 | 83.04 |
| 20 | 1.468 | 86.76 |

## C.3 Results on Novelty, Uniqueness

We report additional metrics on the novelty, uniqueness of generated molecules in Table 14 following previous literature [5, 23, 58]. Novelty measures the percentage of generated molecules not seen in the training set. Uniqueness measures the proportion of non-isomorphic graphs within valid molecules. Higher values indicate better quality for both metrics. The result shows that TACS generates molecules with decreased novelty. This shows that TACS is effective in making generated molecules that stick to the original data distribution while satisfying to meet the target condition.

Table 14: Novelty and uniqueness of generated sample for the target property $\epsilon_{\text{LUMO}}$.

| Method | Novelty (%) | Uniqueness (%) |
|--------|-------------|----------------|
| EDM | 84.5 | 99.9 |
| EEGSDE | 84.8 | 84.8 |
| TACS | 71.6 | 99.8 |

## D  Mathematical Derivations

In our derivation of equivariance properties of the time predictor, we closely adhere to the formal procedures outlined in [23, 57, 47].

**Definition 1** (E(3) Equivariance). *A function $f : \mathbb{R}^{N \times 3} \to \mathbb{R}^{N \times d}$ is E(3)-equivariant if for any orthogonal matrix $\mathbf{R} \in \mathbb{R}^{3 \times 3}$ and translation vector $\mathbf{v} \in \mathbb{R}^3$,*

$$f(\mathbf{R}\mathbf{X} + \mathbf{v}\mathbf{1}^\top) = \mathbf{R} \cdot f(\mathbf{X}), \tag{17}$$

*where $\mathbf{X} \in \mathbb{R}^{N \times 3}$ and $\mathbf{1} \in \mathbb{R}^N$ is the all-ones vector.*

**Definition 2** (Permutation Invariance). *A function $g : \mathbb{R}^{N \times d} \to \mathbb{R}^d$ is permutation-invariant if for any permutation matrix $\mathbf{P} \in \{0, 1\}^{N \times N}$,*

$$g(\mathbf{P}\mathbf{H}) = g(\mathbf{H}), \tag{18}$$

*where $\mathbf{H} \in \mathbb{R}^{N \times d}$.*

**Proposition 1.** *Let $f : \mathbb{R}^{N \times 3} \to \mathbb{R}^{N \times d}$ be an E(3)-equivariant function and $g : \mathbb{R}^{N \times d} \to \mathbb{R}^d$ be a permutation-invariant function. Then, the composition $h = g \circ f : \mathbb{R}^{N \times 3} \to \mathbb{R}^d$ is invariant to E(3) transformations, i.e.,*

$$h(\mathbf{R}\mathbf{X} + \mathbf{v}\mathbf{1}^\top) = h(\mathbf{X}). \tag{19}$$

*Proof.* For any orthogonal matrix $\mathbf{R} \in \mathbb{R}^{3 \times 3}$ and translation vector $\mathbf{v} \in \mathbb{R}^3$,

$$\begin{aligned}
h(\mathbf{R}\mathbf{X} + \mathbf{v}\mathbf{1}^\top) &= g(f(\mathbf{R}\mathbf{X} + \mathbf{v}\mathbf{1}^\top)) \\
&= g(\mathbf{R} \cdot f(\mathbf{X})) \quad \text{(E(3) equivariance of } f) \\
&= g(f(\mathbf{X})) \quad \text{(Permutation invariance of } g) \\
&= h(\mathbf{X}). 
\end{aligned} \tag{20}$$

$\square$

**Theorem 1** (Time Predictor Equivariance). *Let $\mathcal{G} = (\mathcal{V}, \mathcal{E})$ be a graph representing a molecule, where $\mathcal{V} = \{1, \ldots, N\}$ is the set of nodes (atoms) and $\mathcal{E} \subseteq \mathcal{V} \times \mathcal{V}$ is the set of edges (bonds). Let $\mathbf{X}_t \in \mathbb{R}^{N \times 3}$ denote the atomic coordinates and $\mathbf{c} \in \mathbb{R}^d$ be a condition vector at diffusion timestep $t$. Consider a time predictor $p_\phi(t|\mathbf{X}_t, \mathbf{c}) = \text{softmax}(f_\phi(\mathbf{X}_t, \mathbf{c}))$ parameterized by a composition of an E(3)-equivariant graph neural network $EGNN_\phi : \mathbb{R}^{N \times 3} \times \mathbb{R}^{N \times d_h} \times \mathbb{R}^d \to \mathbb{R}^{N \times d'}$, a permutation-invariant readout function $\rho : \mathbb{R}^{N \times d'} \to \mathbb{R}^{d'}$, and a multilayer perceptron $\psi : \mathbb{R}^{d'} \to \mathbb{R}^T$, i.e.,*

$$f_\phi(\mathbf{X}_t, \mathbf{c}) = \psi(\rho(EGNN_\phi(\mathbf{X}_t, \mathbf{H}_t, \mathbf{c}))), \tag{21}$$

*where $\mathbf{H}_t \in \mathbb{R}^{N \times d_h}$ are node features and $T$ is the total number of diffusion timesteps. Then, the time predictor $p_\phi(t|\mathbf{X}_t, \mathbf{c})$ is invariant to E(3) transformations, i.e., for any orthogonal matrix $\mathbf{R} \in \mathbb{R}^{3 \times 3}$ and translation vector $\mathbf{v} \in \mathbb{R}^3$,*

$$p_\phi(t|\mathbf{R}\mathbf{X}_t + \mathbf{v}\mathbf{1}^\top, \mathbf{c}) = p_\phi(t|\mathbf{X}_t, \mathbf{c}), \quad \forall t \in \{1, \ldots, T\}. \tag{22}$$

*Proof.* The E(3)-equivariant graph neural network $\text{EGNN}_\phi$ satisfies [47]:

$$\text{EGNN}_\phi(\mathbf{R}\mathbf{X}_t + \mathbf{v}\mathbf{1}^\top, \mathbf{H}_t, \mathbf{c}) = \mathbf{R} \cdot \text{EGNN}_\phi(\mathbf{X}_t, \mathbf{H}_t, \mathbf{c}). \tag{23}$$

By the permutation invariance of $\rho$ and the invariance of $\psi$ to orthogonal transformations, we have:

$$\begin{aligned}
f_\phi(\mathbf{R}\mathbf{X}_t + \mathbf{v}\mathbf{1}^\top, \mathbf{c}) &= \psi(\rho(\text{EGNN}_\phi(\mathbf{R}\mathbf{X}_t + \mathbf{v}\mathbf{1}^\top, \mathbf{H}_t, \mathbf{c}))) \\
&= \psi(\rho(\mathbf{R} \cdot \text{EGNN}_\phi(\mathbf{X}_t, \mathbf{H}_t, \mathbf{c}))) \\
&= \psi(\rho(\text{EGNN}_\phi(\mathbf{X}_t, \mathbf{H}_t, \mathbf{c}))) \\
&= f_\phi(\mathbf{X}_t, \mathbf{c}).
\end{aligned} \tag{24}$$

Consequently,

$$\begin{aligned}
p_\phi(t|\mathbf{R}\mathbf{X}_t + \mathbf{v}\mathbf{1}^\top, \mathbf{c}) &= \text{softmax}(f_\phi(\mathbf{R}\mathbf{X}_t + \mathbf{v}\mathbf{1}^\top, \mathbf{c})) \\
&= \text{softmax}(f_\phi(\mathbf{X}_t, \mathbf{c})) \\
&= p_\phi(t|\mathbf{X}_t, \mathbf{c}).
\end{aligned} \tag{25}$$

$\square$

# E  Comparison with other related works

While with different motivations and methods, here, we list some of the relevant works and compare their algorithms with ours.

**Comparison with DMCMC**  DMCMC [28] trains a classifier to predict noise levels of the given data during the reverse diffusion process which is similar to our time predictor. However, they use the classifier to estimate the current noise state when conducting MCMC on the product space of the data and the noise. In contrast, our time predictor directly predicts timesteps for correction in diffusion sampling itself. Moreover, while the purpose of noise prediction in DMCMC is for fast sampling, our work use time predictor to accurately produce samples from the desired data distribution.

**Comparison with TS-DPM**  Time-Shift Sampler [34] targets to reduce exposure bias targets similar approach of fixing timesteps during the inference as our time correction method. However, while our method directly selects timesteps based on the time predictor, which has demonstrated robustness in our experiments [34] selects timesteps by calculating variance of image pixels at each step and matching the noise level from the predefined noise schedule which is often inaccurate and expensive. Moreover corrected timestep in [34] is used directly for the start of the next step, while our approach maintains the predefined timestep after accurately estimate clean sample by corrected time step (line 9 in Algorithm 1). By taking every single diffusion step while carefully using predicted time, TACS / TCS can generate samples closer to the target distribution.

To further validate our approach, we provide additional experimental results comparing TS-DDPM [34] and TACS on the QM9 dataset with step size 10. The result in Table 15 shows consistent improvements across various quantum properties which shows the robustness of our approach.

Table 15: Comparison between TACS and Time shift sampler on conditional molecular generation.

| Method | TS-DDPM | | TACS (ours) | |
|---|---|---|---|---|
| | MAE | MS (%) | MAE | MS (%) |
| $C_v$ | 1.066 | 74.89 | 0.659 | 83.6 |
| $\mu$ | 1.166 | 73.55 | 0.387 | 83.3 |
| $\alpha$ | 2.777 | 75.20 | 1.44 | 86.0 |
| $\Delta(\epsilon)$ | 665.3 | 82.72 | 332 | 88.8 |
| HOMO | 371.8 | 72.74 | 168 | 87.3 |
| LUMO | 607.6 | 74.98 | 289 | 82.7 |

# F   Visualization of generated molecules

**Conditional generation with target quantum chemical property**

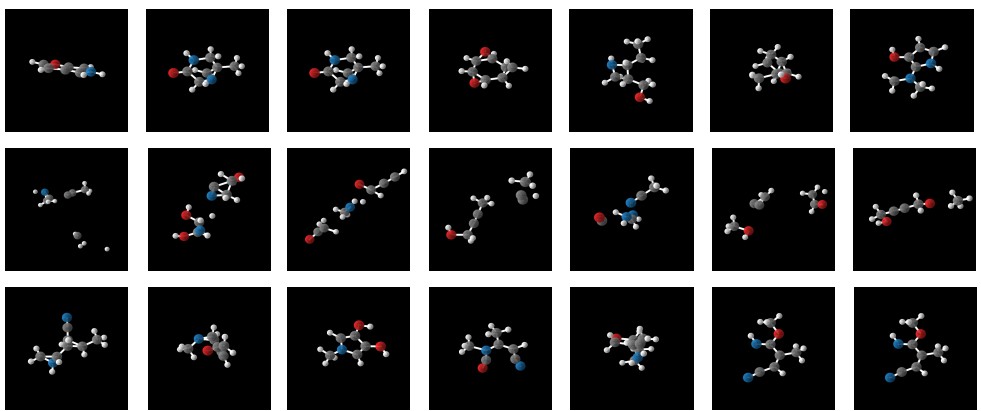

Figure 4: Conditional generation of target property $\alpha$ on QM9. Visualization of molecules generated by TCS (top), online guidance (middle), and TACS (bottom).

**Conditional generation with target structure**

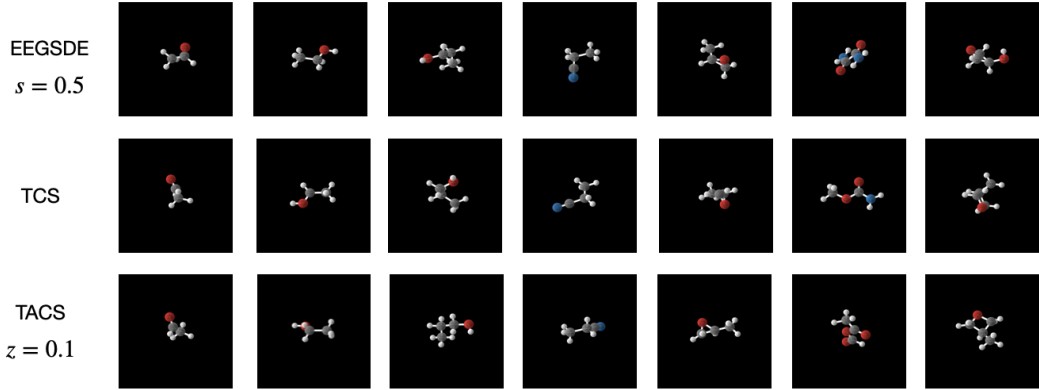

Figure 5: Visualization of how generated molecules align with target structures.

**Unconditional generation on Geom-Drug**

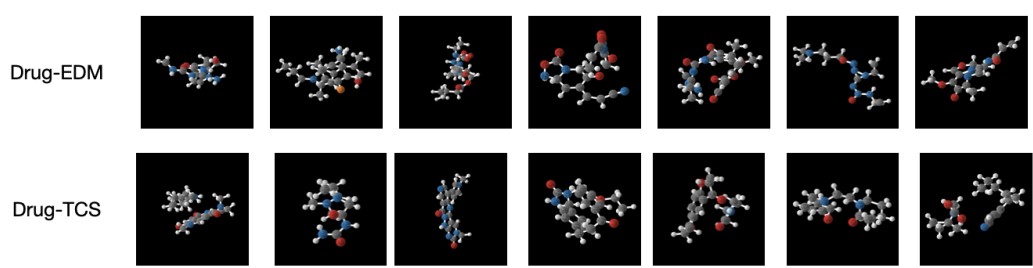

Figure 6: Selection of samples generated by the denoising process of EDM and TCS in Geom-Drug.

