# OpenReview forum: "Conditional Synthesis of 3D Molecules with Time Correction Sampler"
_NeurIPS.cc/2024/Conference — NeurIPS 2024 poster_

### Official Review · Reviewer_ByqK · 2024-07-11

**Soundness:** 3
**Presentation:** 2
**Contribution:** 3
**Rating:** 5
**Confidence:** 3

**Summary:**

This paper focuses on diffusion-model-based molecular inverse design (i.e., conditional molecular generation), and proposes a novel approach to address the inconsistency between target distribution and that after online guidance. Specifically, a time predictor is trained to predict the time of the manifold that the sample lies on, and then a correction is done to ensure that the generated molecules remain on the correct manifold by Tweedie's formula. The proposed method shows satisfactory results on the QM9 dataset.

**Strengths:**

- The motivation is clear and the idea is novel. The inconsistency is indeed a challenge in online guidance of diffusion models. TCS and TCAS solve this problem elegantly.
- The authors have introduced extensive related work to help the audience to better understand the literature.
- The authors provided necessary ablation studies in the appendix.

**Weaknesses:**

- The experiments are not sufficient. More experiments (like designing molecules with given substructures in EEGSDE).
- The idea of TCS is good, and an experiment beyond molecules may further demonstrate the generalizability of the proposed approach. For example, a comparison with [1] is recommended, which the authors have also mentioned when introducing the exposure bias in diffusion models. Besides, I think the accuracy of the time predictor is highly dependent on the data itself. It is suspected that this approach may not work in other fields.
- The presentation of the paper needs to be improved. There are some obvious typos (e.g., line 63). This is just a kind reminder and does not impact my rating of this paper.

[1] Ning, M., Li, M., Su, J., Salah, A.A. and Ertugrul, I.O., 2023. Elucidating the exposure bias in diffusion models. arXiv preprint arXiv:2308.15321.

**Questions:**

See the above weaknesses.

Besides,
- as this paper focuses on the online guidance of the diffusion models for continuous variables. What about the version of TCS and TACS for the discrete variables, if applicable?
- EEGSDE and some other related works that optimize the properties of generated molecules (e.g., [1]) cannot optimize the number of atoms, which is also important in molecule design. Do the authors have some idea about this? (This is only a related question and does not impact my rating.)

[1] Zhou, Xiangxin, Liang Wang, and Yichi Zhou. "Stabilizing Policy Gradients for Stochastic Differential Equations via Consistency with Perturbation Process." arXiv preprint arXiv:2403.04154 (2024).

**Limitations:**

The authors have discussed the limitations in terms of how the signal of guidance is computed. However, I wonder whether the use of computational quantum chemistry methods works as the authors mentioned. My concerns lie on the differentiability required by guidance.

---

> ### Author Rebuttal · Authors · 2024-08-07
>
> Thank you for the constructive and helpful comments. We have addressed your comments below. We appreciate your positive comments that
>
> - Clear motivation and a novel idea to the challenging problem.
> - Extensive related works for the audience.
> - Necessary ablation studies.
>
>
> We initially address your concerns below.
>
> ---
>
> **W1**  Additional experiments on another task.
>
> Thank you for the suggestion. In Table. R2 of the uploaded pdf, we provide the results of our methods (TCS/TACS) in target substructure generation following the setting of EEGSDE and in another dataset GEOM-DRUG (Table. R3). TACS outperforms EEGSDE in similarity scores while maintaining high stability, and achieving higher stability and validity outperforming baselines even in a more complex dataset (GEOM-DRUG). For a detailed analysis of these results, please refer to the discussion in [GR1].
>
> ---
>
>
> **W2**  Generalizability beyond molecules / comparison with [1]
>
> We appreciate your insightful suggestion. As detailed in [GR1], we have applied TCS to image generation on CIFAR-10 dataset and our method results in improved FID scores.
> Our results indicate that TCS effectively mitigates exposure bias across different domains. For instance, on CIFAR-10 dataset, applying TCS only improved the FID score from 3.353 to 2.663, while ADM-ES+TCS further reduced it to 2.179. This consistent improvement across molecular and image domains suggests that our approach addresses fundamental aspects of exposure bias in diffusion models, rather than being data-specific.
> Methodologically, our approach differs from Epsilon Scaling [1] in that we focus on correcting the timesteps based on predicted variance by time predictor, while [1] scales the network output directly. This allows for more precise adjustments at each step, potentially leading to better quality outputs across various domains. For a more detailed analysis of these results and their implications, please refer to [GR1].
>
> ---
>
> **W3**  Correction
>
> We sincerely appreciate your careful reading and for bringing this to our attention. We will thoroughly review and correct any typographical errors in the final version of our manuscript.
>
> ---
>
> **Q1** Online guidance for discrete variables
>
> While our work focus on 3D molecules with continuous variables, adapting TACS for discrete variables using zero-order gradient estimation—similar to our approach with synthetic data and VQE—presents an intriguing direction for future research. We will include a discussion on this potential extension in the limitations and future work section of the manuscript.
>
>
> ---
>
> **Q2** Optimizing number of atoms
>
> Exploring variable atom numbers is indeed a promising direction for future research, which we would like to mention in the final manuscript. TACS could potentially be extended to handle variable atom numbers, possibly by incorporating a learnable atom number prediction step in the generation process.
>
> [1] Ning, M., Li, M., Su, J., Salah, A.A. and Ertugrul, I.O., 2023. Elucidating the exposure bias in diffusion models. arXiv preprint arXiv:2308.15321.
> [2] Zhou, Xiangxin, Liang Wang, and Yichi Zhou. "Stabilizing Policy Gradients for Stochastic Differential Equations via Consistency with Perturbation Process." arXiv preprint arXiv:2403.04154 (2024).

---

> > ### Comment · Reviewer_ByqK · 2024-08-13
> >
> > Thanks for your response.
> >
> > I will keep my current score. It would be promising to introduce the optimization of the number of atoms and extend this framework to the case of discrete variables.

---

> ### Author Response · Authors · 2024-08-14
>
> Dear reviewer ByqK
>
> We are pleased if our response has addressed all of your concerns. If you have any further questions, please let us know.
>
> Your careful review and insightful comments on our paper are greatly appreciated. Thank you once again for putting in your valuable time and effort.

---

### Official Review · Reviewer_1ecU · 2024-07-12

**Soundness:** 3
**Presentation:** 2
**Contribution:** 3
**Rating:** 7
**Confidence:** 4

**Summary:**

This study utilizes a predicted time estimator to correct the data manifold during the guided generation of diffusion models for molecules, to mitigate the discrepancy between the forward and reverse distribution. The authors show that by adjusting the noised sample according to predicted time, as opposed to relying on the pre-defined time schedule, the guided generative process remains on the correct manifold, leading to higher generation quality for several conditional generation tasks.

**Strengths:**

This study offers useful insights into reducing the exposure bias of diffusion models using time correction. It also performs comprehensive analysis and ablation studies on the proposed framework. The results could help future works on diffusion models for 3D molecules in enhancing the generative quality with property guidance.

In addition, the authors demonstrate the possibility of using quantum computing (instead of data-driven classifier, which is the common practice) for the online guidance.

**Weaknesses:**

Though the proposed method and most results are solid, some important results and experimental details seem to be missing or incorrect. See “Questions”. I'm willing to adjust my score if the questions are properly addressed.

**Questions:**

Major:
1\. Line 173: the value of time window size is an important hyperparameter, but the value is not provided. Furthermore, the impact of the time window size choice should be evaluated in the ablation.

2\. Appendix C seems incomplete. How is the function incorporated into the guidance?

3\. Some highlighted values in Table 1 are not the best value, and some values seem abnormally high or low. Please confirm.

4\. A previous work [1] directly searches the time window for a better match of the time-dependent variance. How does the proposed method compare to it?

[1] Li, Mingxiao, et al. "Alleviating exposure bias in diffusion models through sampling with shifted time steps." arXiv preprint arXiv:2305.15583 (2023).

Minor:

1\. Several references to the Appendix needs to be fixed, e.g.:

Line 264 should be B.4;

Line 149: the said comparison is not in the Appendix.

Line 248: the reference is broken and the said result is not in the Appendix.

2\. According to Fig 4, the corrected time becomes very close to the actual time after 400. Appendix B.4 shows the molecule generation quality is the highest if the time correction and OG starts at 400.  Is there any possible relation between these observations?

**Limitations:**

The authors have properly addressed the limitations.

---

> ### Author Rebuttal · Authors · 2024-08-07
>
> Thank you for the constructive and helpful comments. We have addressed your comments below. We appreciate your positive comments that
>
> - Offers useful insights in reducing the exposure bias of diffusion models.
> - Comprehensive analysis and ablation studies which could help future works.
> - Demonstrates the possibility of quantum computing.
>
>
> We initially address your concerns below.
>
> ---
>
> **Q1** Size of Time Window
>
> We appreciate your feedback and we agree that sensitive analysis on time window size can be an important point. We used $\Delta = 10$ for all of  the experimental results reported in our original manuscript and the value is chosen from selecting the best performed MAE values. Here, we provide the experimental results for target property $\alpha$ in Table R6 of the uploaded pdf. The results indicate that performance is relatively stable for $\Delta$ between 2 and 16.
>
> We appreciate this constructive suggestion and we will include the detailed ablation study on window size in our final manuscript for a clearer understanding of our method's behavior.
>
> ---
>
> **Q2** Online guidance using VQE.
>
> In each denoising step of diffusion model, we first apply Tweedie’s formula to estimate clean molecule and use VQE calculates ground state energy of this predicted molecule by updating $\theta$ for $E(x,\theta) = \langle\psi(\theta)|H(x)|\psi(\theta)\rangle$. Then we use the zeroth-order method with respect to the position of atoms $x$ to obtain gradient $\nabla_x E(x, \theta)$. This gradient is used as an online guidance for our synthetic experiment. We will add this information in the Appendix in our final version of the paper.
>
> ---
>
> **Q3**  Table 1 correction.
>
> Thank you for the feedback. Our original intention was to highlight the best performance in conditioning (MAE) with molecular stability above 80%, aligning with our research goal of achieving desired quantum properties while maintaining molecular stability and validity. This is because molecules with lower stability are unlikely to exist in the real world (please refer to Figure 3, second row, in our manuscript for an example). However, we admit that this can cause the confusion of the readers and thus decided to change the notations accordingly.
>
> In the updated table, we use bold for the best overall performance and color highlighting for the best performance with stability above 80%. We believe that this better represents the trade-offs in our method and baselines, while emphasizing the importance of molecular stability in practical applications.
>
>
> ---
>
> **Q4** Comparison with Time Shift Sampler in [1]
>
> Regarding the comparison with [1], we would like to clarify the key differences between TS [1] and our TCS / TACS:
>
> - Our method directly selects timesteps based on a trained time predictor, which has demonstrated robustness in our experiments. In contrast, [1] selects timesteps by calculating variance of image pixels at each step and matching the noise level from the predefined noise schedule.
>
> - In [1], the corrected timestep is used directly for the start of the next step, while our approach maintains the predefined timestep after accurately estimate clean sample by corrected time step. By taking every single diffusion step while carefully using predicted time, TCS can generate samples closer to the target distribution.
>
> To further validate our approach, we provide additional experimental results comparing TS-DDPM [1] and TACS on the QM9 dataset with step size 10 In Table A.3 below.  The consistent improvements across various quantum properties underscore the robustness of our approach.
>
> [Table A.3] Comparison between TS ([1]) and TCS / TACS
>
> | Property | TS MAE | TS Stab. (%) | TACS(Ours) MAE | TACS(Ours) Stab. |
> |----------|--------|--------------|----------------|-------------------|
> | Cv       | 1.066  | 74.89        | 0.659          | 83.6              |
> | μ        | 1.166  | 73.55        | 0.387          | 83.3              |
> | α        | 2.777  | 75.2         | 1.44           | 86.0              |
> | Δε       | 665.3  | 82.72        | 332            | 88.8              |
> | HOMO     | 371.8  | 72.74        | 168            | 87.3              |
> | LUMO     | 607.6  | 74.98        | 289            | 82.7              |
>
>
> We will include this comparison in our final manuscript..
>
> ---
>
> **Q5**  Addition and corrections of Appendix.
>
> We appreciate your careful review, which has helped improve the clarity and completeness of our paper. In our final version, we will change the links and add information accordingly. Please refer to [GR2] for more details on these additions.
>
> ---
>
> **Q6**  Possible relationship between start of OG and Figure 4.
>
> Thank you for the insightful comment. We also think there might be a connection between starting time of online guidance and the convergence of predicted time to the ground truth values as in Fig.4 of the main paper. We suspect this happens because applying online guidance can be especially effective after the sample converges to some degree as pointed out in [2]. Investigating the exact relationship between this convergence and where to start OG would be an interesting future direction.
>
> [1] Li et al., Alleviating exposure bias in diffusion models through sampling with shifted time steps, arXiv 2023
> [2] Han, Xu, et al. "Training-free Multi-objective Diffusion Model for 3D Molecule Generation." The Twelfth International Conference on Learning Representations. 2023.

---

> > ### Comment · Reviewer_1ecU · 2024-08-12
> >
> > I appreciate the authors' detailed response and extensive experiments. I will raise my score to 7.

---

> ### Author Response · Authors · 2024-08-14
>
> Dear Reviewer 1ecU,
>
> We are glad to hear that our response properly addressed your concerns. Your careful review and insightful comments on our paper are greatly appreciated. Thank you once again for putting in your valuable time and effort.

---

### Official Review · Reviewer_9Yep · 2024-07-13

**Soundness:** 3
**Presentation:** 3
**Contribution:** 3
**Rating:** 8
**Confidence:** 4

**Summary:**

This paper presents a framework for generating 3D molecules called Time-Aware Conditional Synthesis TACS. The proposed approach uses conditional generation with adaptively controlled plug-and-play online guidance into a diffusion model to drive samples toward the desired properties while maintaining validity and stability. To prevent generated samples deviating from the data distribution during the conditional generation process authors introduce a Time Correction Sampler to control guidance and ensure that the generated molecules remain on the correct manifold at each reverse step of the diffusion process. Authors compare their TACS results with Equivariant Diffusion Models EDM and Equivariant Energy Guided Stochastic Differential Equations EEGSDE.

**Strengths:**

This paper presents a framework for generating 3D molecules called Time-Aware Conditional Synthesis TACS. The proposed approach uses conditional generation with adaptively controlled plug-and-play online guidance into a diffusion model to drive samples toward the desired properties while maintaining validity and stability. To prevent generated samples deviating from the data distribution during the conditional generation process authors introduce a Time Correction Sampler to control guidance and ensure that the generated molecules remain on the correct manifold at each reverse step of the diffusion process. Authors compare their TACS results with Equivariant Diffusion Models EDM and Equivariant Energy Guided Stochastic Differential Equations EEGSDE.

**Weaknesses:**

How to efficiently use the Time Corrected Sampler and whether this method improves the performance in other domains such as in image generation.

**Questions:**

Have the authors experimented TACS approach on other datasets?

**Limitations:**

In this study, authors have used a trained neural network to estimate chemical properties of each molecule. Using an exact computational chemistry-based method could improve the guidance.

---

> ### Author Rebuttal · Authors · 2024-08-07
>
> Thank you for the constructive and helpful comments. We have addressed your comments below.
>
> **W1**  Generalizability of TCS
>
> As detailed in [GR1], we have applied TCS to image generation using the CIFAR-10 dataset, demonstrating significant improvements in FID scores. These results show that our method generalizes well beyond molecular data. We believe that our algorithm can be applicable in other domains as well since the design of TCS and TACS does not include any domain-specific knowledge. It would be an excellent direction for future work to investigate our method in various domains.
>
> **Q1** Experiments on other datasets
> As mentioned in [GR1], we have conducted experiments on the Geom-Drug dataset for molecule generation. Our method shows improved performance in generating molecules with molecular stability compared to the previous baselines like in EEGSDE. This further demonstrates TACS's ability to generalize to more complex molecular datasets and tasks. We will add these results to the final version of our manuscript.

---

### Official Review · Reviewer_VutT · 2024-07-13

**Soundness:** 3
**Presentation:** 2
**Contribution:** 3
**Rating:** 6
**Confidence:** 4

**Summary:**

This paper proposes Time-Aware Conditional Synthesis (TACS), a method that aims to improve the robustness of property-conditioned diffusion models for 3D molecule generation. The key idea is to mitigate the exposure bias of the conditional denoising process by training a time prediction model that matches samples to the most likely marginal distribution of the forward process before applying online guidance via Tweedie's formula. Experiments on a synthetic dataset and QM9 show that TACS generates valid samples that match the conditioning label more closely than alternative methods.

**Strengths:**

* The paper introduces a promising approach to keep the generated samples aligned with the marginal distributions of the forward process and address the problem of exposure drift in conditional molecule generation.
* The paper demonstrates that TACS performs better than several well-established baselines on the QM9 dataset, showing improvements in generating molecules with desired quantum properties while maintaining stability and validity.
* The paper is well-written and clearly outlines the methodology and motivation behind the method.

**Weaknesses:**

* The method is only compared to well-established baselines on a single dataset (QM9). It would be good to evaluate the model on at least one other benchmark, to ensure that the results indicate a general trend and are not specific to the very distinct data distribution of QM9.
* None of the quantitative empirical results in Section 5 include error bars or measures of statistical significance.
* The main text contains multiple references to Appendix A.1, which I assume are incorrect links. I could not find the comparison with relevant work in [2] (referenced in line 149) or any details on the MAE distribution of samples below and above an 80% stability threshold (referenced in lines 227-229). Furthermore, the model performance analysis for $m>1$ MCMC samples referred to in line 248 seems to be missing from the Appendix.
* Minor Point: The results in Table 1 and Figure 4 show that online guidance is often able to generate samples with much lower property MAEs at the expense of stability and validity. This tradeoff is discussed in the Ablation Studies paragraph, but it would be good to calibrate the claim that TACS "outperforms competitive baseline methods across all metrics", since online guidance refers to a published baseline model [1].

**Questions:**

* Do the stability values reported in Table 1 refer to atom or molecule stability?
* The online guidance results in Table 1 are sometimes much worse than those reported in [1], especially for $\mu(D)$, $\epsilon_\text{HOMO}$ and $\epsilon_\text{HOMO}$. Do you know what could cause that?
* The paper mentions having to use a classification rather than a regression model to estimate the time step $t_\text{pred}$ because $p(t\vert\mathbf{x})$ cannot be estimated from a point estimate (line 147). However, Algorithm 1 then discards the information about the full distribution by taking the maximum likelihood estimate (I assume that's what $\operatorname{argmax}\phi(\mathbf{x}')$ in line 6 means). Would it be better to use $\mathbb{E}_{p(t\vert\mathbf{x}')}[t]$ instead?
* Is the time predictor is only trained on samples from the forward process?

---

[1] Han, Xu, et al. "Training-free Multi-objective Diffusion Model for 3D Molecule Generation." The Twelfth International Conference on Learning Representations. 2023.
[2] Kim, Beomsu, and Jong Chul Ye. "Denoising mcmc for accelerating diffusion-based generative models." arXiv preprint arXiv:2209.14593 (2022).

**Limitations:**

The authors list the reliance on a potentially flawed predictive model for sample ranking as the main limitation. It would be good to also discuss any limitations of the method itself.

---

> ### Author Rebuttal · Authors · 2024-08-07
>
> Thank you for the constructive and helpful comments. We have addressed your comments below. We appreciate your positive comments that our work
>
> - Introduces a promising approach to the exposure bias
> - Show improvements compared to baselines with desired quantum properties while maintaining stability and validity.
> - Well-written and clearly outlines the methodology and motivation.
>
> We initially address your concerns below.
>
> ---
>
> **W1** Evaluation on other benchmarks
>
> As detailed in General Response-[GR1], we provide additional experiments on Geom-Drug and CIFAR-10 which demonstrate TACS's ability to generalize beyond QM9, addressing molecules with more complex structures (Geom-Drug) and even extending to image generation (CIFAR-10).
>
> [Table A.1] TCS on CIFAR 10
>
> |     | ADM   | ADM-ES |ADM-TCS(Ours)| ADM-ES+TCS(Ours)|
> |-----|-------|--------|-------------|-----------------|
> | FID | 3.353 | 2.213  | 2.663       |     2.179       |
> |     |       |        |             |                 |
>
> ---
>
> **W2**  Error bars and statistical significance
>
> Thank you for pointing out the missing error bars. We have included them in Table.R1 of the uploaded pdf and will add in our final manuscript.
>
> ---
>
> **W3** Missing information in Appendix A.
>
> We appreciate your careful review. In our final version, we will expand on several points as follows:
>
> **R3-1** Comparison with DMCMC [3]
> DMCMC uses a classifier to predict noise levels for MCMC on the product space of data. In contrast, our time predictor directly predicts timesteps for correction in diffusion sampling itself. Moreover, while the purpose of noise prediction in [3] is for fast sampling, our work use time predictor to accurately produce samples from the desired data distribution.
>
> **R3-2** Impact of MC numbers
> We will also include detailed analysis on MC numbers (line 248), demonstrating TACS's robust performance across different sample sizes. We will also include our analysis of experiments conducted on the effect of MC sample numbers. Unlike [1], where increasing samples from 5 to 10 more than doubled performance at increased computational cost, our method shows robust performance across different sample numbers.
>
> For details on these additions, we kindly refer to [GR2].
>
> **R3-3** MAE distribution given the threshold.
> We will also clarify our discussion on MAE and molecular stability (MS) thresholds. The revised text will emphasize TACS's lower MAE compared to methods achieving MS above 80%, and its performance even for samples below this threshold.
>
> ---
>
>
> **W4** Calibrate the claim that “TACS outperforms competitive baseline methods across all metrics", since online guidance refers to a published baseline model [1].
>
> We will revise our claim as "TACS outperforms competitive baselines when considering MAE, stability, and validity collectively," which more accurately reflects our results. The updated Table. R1 of the uploaded pdf in the supplementary material supports this claim.
>
>
> ---
>
> **Q1**  Stability values reported in Table 1.
>
> Stability values denote the molecule stability.
>
>
>
> **Q2** Online guidance results in Table 1.
>
> Online guidance yields higher MAE for some properties such as $\mu$ compared to that reported in [1], and we suspect this stems from the sensitivity of online guidance strength.
>
> First, data distribution of $\mu$ in the QM9 training dataset is sharp as can be seen in Figure.6-(a) in [2] the distribution of $\mu$.
>
> This in turn, can make online guidance sensitive to its guidance strength, where we sometimes observe diverging gradients when naively applying online guidance.
>
> Additionally, we provide additional experimental results for impact of online guidance strength on $\mu$ to support our claim.
>
> We will include the above result in our final version of the paper for the better understanding of the reader.
>
> **Q3** How to obtain predicted time from the time predictor
>
> Thank you for an interesting question. We provide experimental results on expectation-based $\mathbb{E}[{\phi(\boldsymbol{x})}]$ and argmax-based time prediction in the table below. The result shows that the performance remains robust in general with some trade-offs, where argmax-based prediction achieves slightly better MAE while the expectation-based approach maintains higher molecular stability.
>
> [Table A.2] Different type of time prediction. Above is MAE for target condition and below is molecular stability of generated samples.
> | Method     | Cv     | $\mu$      | $\alpha$   | $\Delta(\epsilon)$   | $\epsilon_{HOMO}$ | $\epsilon_{LUMO}$ |
> |------------|--------|--------|--------|-------|-------|-------|
> | Expectation| 0.7032 | 0.4511 | 1.559  | 351.7 | 181.8 | 334.3 |
> | Argmax     | 0.659  | 0.387  | 1.44   | 332   | 168   | 289.0 |
>
> | Method     | Cv     | $\mu$      | $\alpha$   | $\Delta(\epsilon)$    | $\epsilon_{HOMO}$ | $\epsilon_{LUMO}$ |
> |------------|--------|--------|--------|-------|-------|-------|
> | Expectation| 84.67 | 90.18 | 87.3  | 90.71 | 89.84 | 90.11 |
> | Argmax     | 83.3  | 83.3  | 86.0  | 88.8  | 87.3  | 82.7  |
>
> ---
>
> **Q4**  Is the time predictor only trained on samples from the forward process?
>
> Yes, we trained the time predictor with the samples with the forward process as described in Line 142. Our work used a time predictor for reducing the gap between the marginal distribution of the forward and reverse processes, and thereby it provides good guidance to the desired data distribution only when trained with noisy samples from the forward process.
>
> ---
>
> [1] Han, Xu, et al. "Training-free Multi-objective Diffusion Model for 3D Molecule Generation." The Twelfth International Conference on Learning Representations. 2023.
> [2] Bao, Fan, et al. "Equivariant energy-guided sde for inverse molecular design." The eleventh international conference on learning representations. 2022.
> [3] Kim, Beomsu, and Jong Chul Ye. "Denoising mcmc for accelerating diffusion-based generative models." arXiv preprint arXiv:2209.14593 (2022).

---

> > ### Comment · Reviewer_VutT · 2024-08-12
> >
> > I would like to thank the authors for the detailed response. I believe that the additional experimental results for the GEOM-DRUG and CIFAR-10 datasets corroborate the reported performance gains on the QM9 dataset and strengthen the experimental section of the paper. I will update my score accordingly.

---

> > > ### Author Response · Authors · 2024-08-14
> > >
> > > Dear reviewer VutT
> > >
> > > We are glad to hear that our response properly addressed your concerns. Your careful review and insightful comments on our paper are greatly appreciated. Thank you once again for putting in your valuable time and effort.

---

### Author Rebuttal · Authors · 2024-08-07

General response

Dear reviewers and AC,

We sincerely appreciate your valuable time and effort in reviewing our manuscript. Your insightful feedback has been instrumental in improving our work.

Our paper introduces Time-Aware Conditional Synthesis (TACS), a novel approach for conditional 3D molecular generation using diffusion models. To our best knowledge, this is the first work to systematically achieve the generation of molecules that simultaneously meet desired conditions, stability, and validity. As highlighted by multiple reviewers, TACS offers a principled and effective method (VutT, qUzm, 9Yep, 1ecU), unlike existing works of conditional molecular generation that focus solely on desired properties. Our approach has been thoroughly validated through extensive quantitative and qualitative experiments (VutT, qUzm, 1ecU) and presented comprehensively (VutT, qUzm, ByqK).

We have carefully considered the multiple concerns raised by reviewers and have addressed them comprehensively as follows. We kindly request that you review our responses below, as well as the attached supplementary PDF file.

---

**[GR1] Generalizability Beyond QM9 Dataset (VutT, 9Yep, ByqK)**

To demonstrate the broad applicability of TACS, we conducted additional experiments on:

1. Generating molecules with target structures on QM9 dataset (Different task).
2. The Geom-Drug dataset for molecule generation, demonstrating performance beyond QM9's distinct distribution.(Bigger dataset)
3. Image generation in CIFAR-10 dataset, showcasing generalizability to a different domain.(Other domain)

For different task, we conduct additional experiments on target structure generation as in [1].
We tested our algorithms (TACS) on QM9 dataset and showed that TACS achieves a Tanimoto similarity score of 0.792±0.077 with 90.42% of molecular stability, outperforming the reported and reproduced values of [1] with high margin. This demonstrates TACS's ability to generalize to different tasks. We kindly refer the reader to see the Table. R2 in our uploaded pdf file for more details.

We also tested Time Correction Sampler(TCS) on the Geom-Drug dataset on unconditional generation of 3D molecules. As can be seen in the Table. R3 in the attached pdf file, the result shows improved performance in generating molecules with target structures.

Finally, we tested our algorithm on the image dataset. On CIFAR-10 dataset our Time Correction Sampler (TCS) achieves improved FID scores (3.353 to 2.661) without any fine-tuning of the hyperparameters.The improvement in FID score shows the effectiveness of our time correction approach in mitigating exposure bias, a problem common to diffusion models across various domains.


We believe these results altogether demonstrates that our method generalizes well beyond molecular data, indicate possible applications in other tasks, datasets, and domains.

---

**[GR2] Additional Analyses and Clarifications** (VutT, 1ecU): We appreciate the reviewers' thorough examination of our work. In our final version, we will expand on several points to offer a more comprehensive analysis:

(Line 149) Regarding the DMCMC [2] comparison, we will add a more detailed discussion in the Appendix. This will include key differences such as: DMCMC uses a classifier to predict noise levels for MCMC on the product space of data and noise, focusing on earlier generation stages. In contrast, our time predictor directly predicts timesteps for correction in diffusion sampling, aiming to produce samples closer to the desired data distribution.

(Line 248) We will also include our analysis of experiments conducted on the effect of MC sample numbers. Unlike [3], where increasing samples from 5 to 10 more than doubled performance at increased computational cost, our method shows robust performance across different sample numbers. As shown in Table. R4 of the supplementary material:

| Num. Samples | MAE   | Stab. (%) |
|--------------|-------|-----------|
| 1            | 1.44  | 86.0      |
| 5            | 1.395 | 86.35     |
| 10           | 1.505 | 82.21     |
| 15           | 1.545 | 83.04     |
| 20           | 1.468 | 86.76     |

These results  indicate that TACS maintains consistent performance in terms of both MAE and molecular stability across different numbers of MC samples. This stability eliminates the need for higher m values, avoiding extra computational costs while maintaining accuracy.

All these modifications and additional analyses will be reflected in our final manuscript accordingly.

---

**Reference**


[1] Bao, Fan, et al. "Equivariant energy-guided sde for inverse molecular design." The eleventh international conference on learning representations. 2022.
[2] Kim, Beomsu, and Jong Chul Ye. "Denoising mcmc for accelerating diffusion-based generative models." arXiv preprint arXiv:2209.14593 (2022).
[3] Han, Xu, et al., Training-free Multi-objective Diffusion Model for 3D Molecule Generation, ICLR 2023.

---

### Decision · Program_Chairs · 2024-09-25

**Decision:**

Accept (poster)

**Comment:**

The paper introduces time-correction sampling, an approach to ensure guided samples stay close to the data manifold. Briefly, after a guidance step, a completely denoised step is predicted using Tweedie's formular, this is then noised again using the forward noising process. Rather than using the diffusion time directly, a new diffusion time is predicted by another model, to correct for exposure bias, e.g. mismatch between marginals of the forward and backwards diffusion models. The idea is interesting, and the reviewers all agree with this. The authors address the reviewers' concerns, and while the focus here is on molecular application, I envision this approach having an impact broadly, e.g. in engineering applications. My primary criticism is that I would have loved to see a simple low dimensional illustration of the method to illustrate the method does what the authors claim. While the results suggest this, there is no formal or experimental proof of the approach.